# Time-domain observation of interlayer exciton formation and thermalization in a MoSe₂/WSe₂ heterostructure

Veronica R. Policht [1,6] ✉, Henry Mittenzwey[2] ✉, Oleg Dogadov[1], Manuel Katzer [2], Andrea Villa[1], Qiuyang Li[3], Benjamin Kaiser [4], Aaron M. Ross[1], Francesco Scotognella[1,7], Xiaoyang Zhu[3], Andreas Knorr[2], Malte Selig[2], Giulio Cerullo [1,5] & Stefano Dal Conte[1] ✉

Vertical heterostructures of transition metal dichalcogenides (TMDs) host interlayer excitons with electrons and holes residing in different layers. With respect to their intralayer counterparts, interlayer excitons feature longer lifetimes and diffusion lengths, paving the way for room temperature excitonic optoelectronic devices. The interlayer exciton formation process and its underlying physical mechanisms are largely unexplored. Here we use ultrafast transient absorption spectroscopy with a broadband white-light probe to simultaneously resolve interlayer charge transfer and interlayer exciton formation dynamics in a MoSe₂/WSe₂ heterostructure. We observe an interlayer exciton formation timescale nearly an order of magnitude (~1 ps) longer than the interlayer charge transfer time (~100 fs). Microscopic calculations attribute this relative delay to an interplay of a phonon-assisted interlayer exciton cascade and thermalization, and excitonic wave-function overlap. Our results may explain the efficient photocurrent generation observed in optoelectronic devices based on TMD heterostructures, as the interlayer excitons are able to dissociate during thermalization.

Monolayer (ML) transition metal dichalcogenides (TMDs) exhibit remarkable physical properties which make them an ideal platform to study exciton physics and to realize novel optoelectronic devices[1]. The dimensional reduction from bulk to 2D results in the formation of strongly bound excitons due to increased quantum confinement and reduced Coulomb screening, along with spin-valley locking due to the inversion symmetry breaking[2,3]. The ability to vertically stack multiple TMDs, forming van der Waals heterostructures (HS) without lattice matching constraints, has dramatically increased recent interest in these materials[4]. Despite the fact that weak out-of-plane interactions

largely preserve the electronic structures of each layer, stacked HS display novel properties and functionalities not present in constituent monolayers. HS with Type II band alignment, where the valence band maximum and the conduction band minimum are in different layers, can host interlayer excitons (ILX) (Fig. 1a) which arise following interlayer charge transfer (ICT) (Fig. 1b)[5–7] and consist of spatially separated Coulomb-bound electron-hole states with binding energies up to 100s meV[8]. The ILX in TMD-HS is commonly detected via its photoluminescence (PL) in the near-infrared (NIR), below the energy of the optical gap of the two layers (Fig. 1c)[9–13]. The ILX is characterized by a

[1]Department of Physics, Politecnico di Milano, Piazza Leonardo da Vinci 32, Milano 20133, Italy. [2]Institut für Theoretische Physik, Nichtlineare Optik und Quantenelektronik, Technische Universität Berlin, Hardenbergstraße 36, 10623 Berlin, Germany. [3]Department of Chemistry, Columbia University, 3000 Broadway, New York, NY 10027, USA. [4]Zuse-Institut Berlin, Takustraße 7, 14195 Berlin, Germany. [5]CNR-IFN, Piazza Leonardo da Vinci 32, Milano 20133, Italy. [6]Present address: NRC Postdoc residing at U.S. Naval Research Laboratory, 4555 Overlook Avenue SW, Washington, DC 20375, USA. [7]Present address: Department of Applied Science and Technology, Politecnico di Torino, Corso Duca degli Abruzzi 24, Torino 10129, Italy. ✉e-mail: vpolicht@umich.edu; h.mittenzwey@tu-berlin.de; stefano.dalconte@polimi.it

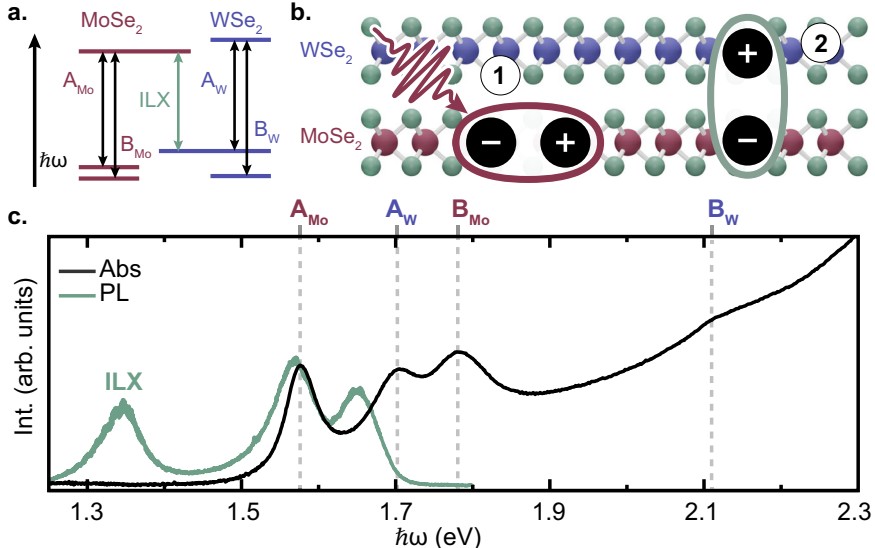

**Fig. 1 | Interlayer Exciton formation in MoSe₂/WSe₂ HS. a** Type II band alignment diagram of the MoSe₂/WSe₂ HS. The relevant intralayer and interlayer exciton transitions are depicted with arrows. **b** ILX formation in the MoSe₂/WSe₂ HS. An intralayer exciton is resonantly excited in the MoSe₂ layer (1). IHT to WSe₂ causes the formation an optically bright ILX (2). **c** Linear absorption (black trace) and PL (green trace) spectra of the HS at 77 K. The PL spectrum was measured in a confocal Raman microscope (inVia, Renishaw) using 530 nm continuous wave excitation at $9 \times 10^7 \mu W/cm^2$. The four peaks in the linear absorption, marked by vertical dashed lines, correspond to A/B intralayer excitons. The PL spectrum reveals a spectral feature below the intralayer exciton absorption edge, which corresponds to ILX emission peak.

long recombination time (up to hundreds of nanoseconds[14,15]) and reduced oscillator strength (two orders of magnitude lower than that of intralayer excitons) due to the small spatial overlap of the electron and hole wave functions and, in some instances, their momentum-indirect character[9,12,16–18]. One of the most intriguing properties of TMD-HS is the lateral confinement of the ILX within moiré potentials, which are formed at small interlayer twist angles[13,19–22].

ILX dynamics have been explored in different TMD-HS by time-resolved techniques[8]. The ILX recombination dynamics measured by time-resolved PL exhibit multiple decay components which exceed the recombination lifetimes of intralayer excitons by several orders of magnitude[9,23]. The ILX relaxation dynamics have also been indirectly inferred from the photobleaching (PB) dynamics of intralayer excitons as measured by transient absorption (TA) optical spectroscopy[24]. On the other hand, the formation dynamics of the ILX are very difficult to access directly due to several factors, including the weak oscillator strength[25,26] and the rapidity of the ICT process that leads to ILX formation, which typically ranges from tens to hundreds of femtoseconds[5–7,24,27,28]. Time-resolved PL lacks the temporal resolution required to observe these processes. Attempts to resolve the transition from intralayer excitons to ILX have been done by measuring the formation of a novel *1s-2p* transition in the mid-IR range[29]. More recently, time- and angle-resolved photoemission spectroscopy (tr-ARPES) experiments have addressed ILX formation dynamics[30,31] by probing the energy-momentum dispersion of photoexcited quasi-particles in real time and distinguishing excitons from single particle states according to their energy-momentum dispersion[31–36]. A recent tr-ARPES study[31] was able to track the ILX formation process following a phonon-assisted interlayer electron transfer as mediated by inter-mediate scattering to the Σ valleys, where Σ has also been referred to as the Q/Λ point[12,37,38]. Despite the strengths of this technique, the limited energy resolution and the extremely low intensity of the photoemission signal above the Fermi level complicates the ability to disentangle ILX formation from ICT dynamics[39].

Here we use ultrafast optical TA spectroscopy to directly probe the transient optical response of the ILX in a MoSe₂/WSe₂ HS. These measurements are enabled by highly stable and broadband white light probe pulses spanning the visible to the NIR. The resulting high signal-to-noise measurements are capable of resolving the ILX formation dynamics through its weak TA signal (two orders of magnitude lower than the TA of the intralayer exciton) while simultaneously measuring intralayer exciton and interlayer hole transfer (IHT) dynamics. The ILX signal shows a delayed growth on a picosecond timescale, which is significantly longer than the experimentally measured 100-fs IHT process from MoSe₂ to WSe₂. We simulate the exciton dynamics by solving the microscopic Heisenberg equations of motion and find that the difference in formation timescales is due to phonon-assisted hole tunneling of photo-excited excitons which gives rise to hot ILX populations that quickly exchange energy and momentum with pho-nons. The relaxation down to the ILX ground state proceeds through multiple scattering processes involving higher energy interlayer *s* states. Our simulations demonstrate that these hot ILX populations contribute strongly to the WSe₂ PB signal but only weakly to the optically bright ILX transition, resulting in the relative delay of the ILX signal as the populations cool.

## Results
### Experimental results
Experiments are performed on a large-area (mm-scale) MoSe₂/WSe₂ HS fabricated using a gold tape exfoliation method (see details in Supplementary Note 1.1)[40]. The HS is prepared with a 4°, or near aligned, interlayer twist angle, characterized using polarization-resolved second harmonic generation (Supplementary Fig. 1a). We focus on the 4° HS in the main manuscript, though experiments were also performed on a nearly anti-aligned MoSe₂/WSe₂ HS with twist angle of 57° (Supplementary Fig. 1b). HS prepared with small twist angles away from either aligned (0°) or anti-aligned (60°) have been shown to have strong ILX PL signals[11]. A clear signature of the ILX in the 4° HS is the low-energy peak in the PL spectrum at $\hbar\omega = 1.35$ eV (Fig. 1c) to the red of the intralayer PL peaks, which are quenched relative to the PL signals of the individual MLs (Supplementary Fig. 7). Both momentum-direct ($K - K$) and momentum-indirect ($\Sigma - K$) electron-

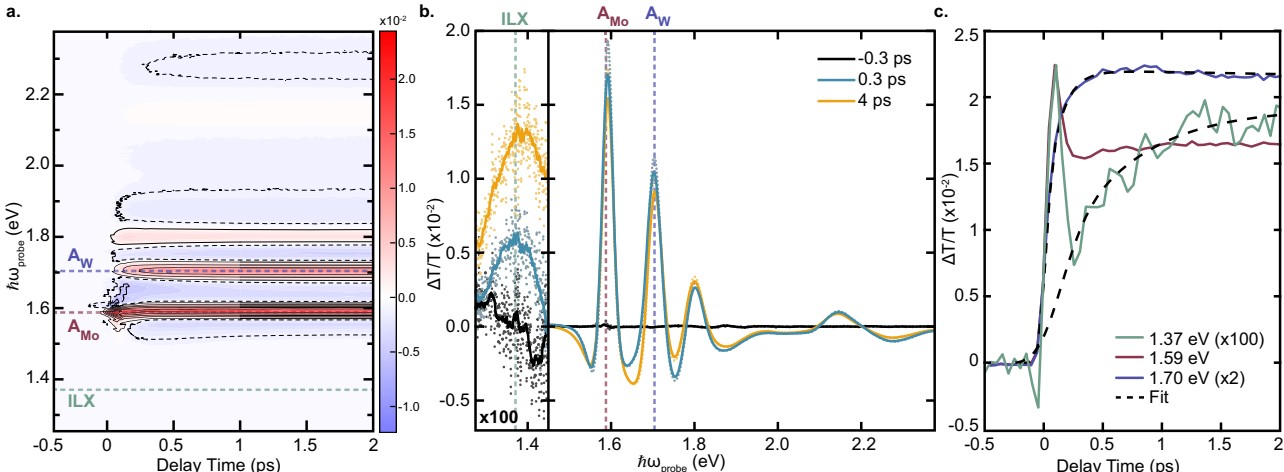

**Fig. 2 | Transient Optical Response of MoSe$_2$/WSe$_2$ HS. a** 2D $\Delta T/T$ map as a function of probe energy and delay time. The map displays positive PB (red) and negative PIA signatures (blue) of the intralayer excitons. Dashed lines indicate the probe energy of the three main peaks of interest at A$_W$ (purple), A$_{Mo}$ (red), and the ILX (green). **b** $\Delta T/T$ spectra from (**a**) at selected early time delays. The region below the intralayer exciton optical gap is multiplied by a factor of 100 to highlight the weak ILX peak in the near IR. Solid lines in the near-IR region are a smoothed representation of the raw data in dots. **c** Temporal dynamics of the A$_{Mo}$ (red, $\hbar\omega$ = 1.59 eV), A$_W$ (purple, $\hbar\omega$ = 1.7 eV), and ILX (green, $\hbar\omega$ = 1.37 eV) in the first 2 ps. The A$_W$ and ILX peaks are multiplied by factors of 2 and 100, respectively, to emphasize the delayed rise of these signatures. Dashed lines are the fits to the data.

hole transitions are expected to contribute to the PL of the ILX[9,12,16,38] which can help to account for the width of the peak in Fig. 1c. We apply ultrafast TA spectroscopy with a broadband white-light continuum (WLC) probe to study the ILX formation dynamics. We selectively photoexcite the MoSe$_2$ layer by tuning a narrow-band (10 nm) 70-fs pump pulse on resonance with the A exciton peak of MoSe$_2$ ($\hbar\omega$ = 1.58 eV) with a fluence of 3 μJ/cm$^2$ (Supplementary Fig. 2a). For simplicity, when discussing the experimental results, we adopt a notation for the intralayer A and B excitons of each layer using a subscript corresponding to the layer's transition metal (A$_X$ and B$_X$ where $X$ = Mo or W). The WLC probe beam is generated in an yttrium aluminum garnet (YAG) crystal pumped by a 1 eV pulse and spans a broad spectral window (1.2–2.3 eV) that includes both intralayer excitons and the ILX of the HS (Supplementary Fig. 2a). The pump and probe are focused to spot diameters on the order of 100 μm, such that sample heterogeneity contributes to inhomogeneous broadening of the TA peaks. Unless otherwise stated, the pump and probe pulses are co-circularly polarized such that they access the ($K - K$) optical transitions.

Figure 2a reports a 2D map of the differential transmission ($\Delta T/T$) signal at 77 K as a function of pump-probe delay and probe photon energy. The spectrum is dominated by positive PB signals (red contours) at the energies of the intralayer excitons of both layers (i.e. A$_{Mo}$, A$_W$, B$_{Mo}$, B$_W$) with signal strengths on the order of $\Delta T/T \sim 10^{-2}$. The transient signal at the energy of A$_{Mo}$ exciton displays an instantaneous (i.e. pulsewidth-limited) build-up and is the result of the interplay of multiple processes, including quenching of the exciton oscillator strength due to phase-space filling, energy renormalization, and lineshape broadening of the exciton resonance due to Coulomb many-body effects[41]. The same formation dynamics are observed for the B$_{Mo}$ resonance as a consequence of the exchange-driven mixing of the excitons in the ($K - K$) valley[42] and light-induced reduction of the Coulomb screening, leading to an instantaneous renormalization of the exciton resonances[43]. Following excitation of the A$_{Mo}$ exciton, the A$_W$ resonance shows a delayed PB due to IHT from MoSe$_2$ to WSe$_2$. The finite build-up time of the A$_W$ PB signal (0.20 ± 0.06 ps, Fig. 2c) provides a direct estimate of the timescale of the hole scattering process, which is in good agreement with previous observations[24]. The B$_W$ excitonic resonance shows the same delayed signal formation owing to coupling with the A$_W$ exciton.

Focusing on the ILX energy region of the 2D $\Delta T/T$ map below the optical gap of the intralayer excitons, the TA spectrum at 4 ps pump-probe delay shows a broad peak with a signal strength $\Delta T/T \sim 10^{-4}$ (Fig. 2b). We confirm that this peak is unique to the HS through control measurements on isolated MoSe$_2$ and WSe$_2$ MLs (see Supplementary Note 2.5) and attribute it to PB of the ILX following IHT. The relative strengths of the TA signals of the intralayer excitons and of the ILX are in good agreement with the two orders of magnitude difference in the static transition dipole moments predicted by theory[44] and measured experimentally[25,26]. Moreover, ILX exhibits valley circular dichroism as reported in Supplementary Note 2.6. We note that the ILX signature is peaked at slightly higher energy in the TA measurements compared to the ILX emission peak in Fig. 1. We attribute this energy mismatch to the fact that the optically bright ILX signal measured via TA is dominated by momentum-direct ($K - K$) transitions[26] compared to the ILX PL signal which includes contributions from momentum-direct and momentum-indirect transitions[12,17]. Figure 2c reports the formation dynamics of the ILX. Compared with the dynamics of A$_{Mo}$ and A$_W$ excitons (red and purple, respectively in Fig. 2c), the ILX TA signal (green) shows a remarkably slower formation timescale (0.8 ± 0.3 ps), significantly longer than the timescale of IHT. This behavior suggests that the optically bright ILX does not form immediately upon IHT and that inferring ILX formation timescales from ICT signatures may be inadequate. The instantaneous peak in the ILX TA signal is due to the weak coherent artifact of the substrate during the pulse overlap, as confirmed by a control measurement on a blank substrate (see Supplementary Note 2.4). This weak coherent signal is not observed at A$_W$ due to its much higher signal strength.

A recent ultrafast TA experiment performed with excitation densities above the exciton Mott transition reported the generation of an interlayer electron-hole plasma whose optical signature consisted of a broad photoinduced absorption (PIA) plateau extending below the optical gap of the HS[14]. The peak we observe at 1.37 eV is not related to this effect because (i) our excitation density ($\approx$3.6 × 10$^{11}$cm$^{-2}$, see Methods for details) is well below the Mott threshold and (ii) the ILX signal shows a positive signal consistent with a PB signature. At higher excitation densities, we find that the ILX PB signal persists while a negative PIA signal, related to pump-induced modification of the A$_{Mo}$ excitonic resonance[43], increases in strength with fluence until it obscures the weak ILX signature (Supplementary Fig. 9).

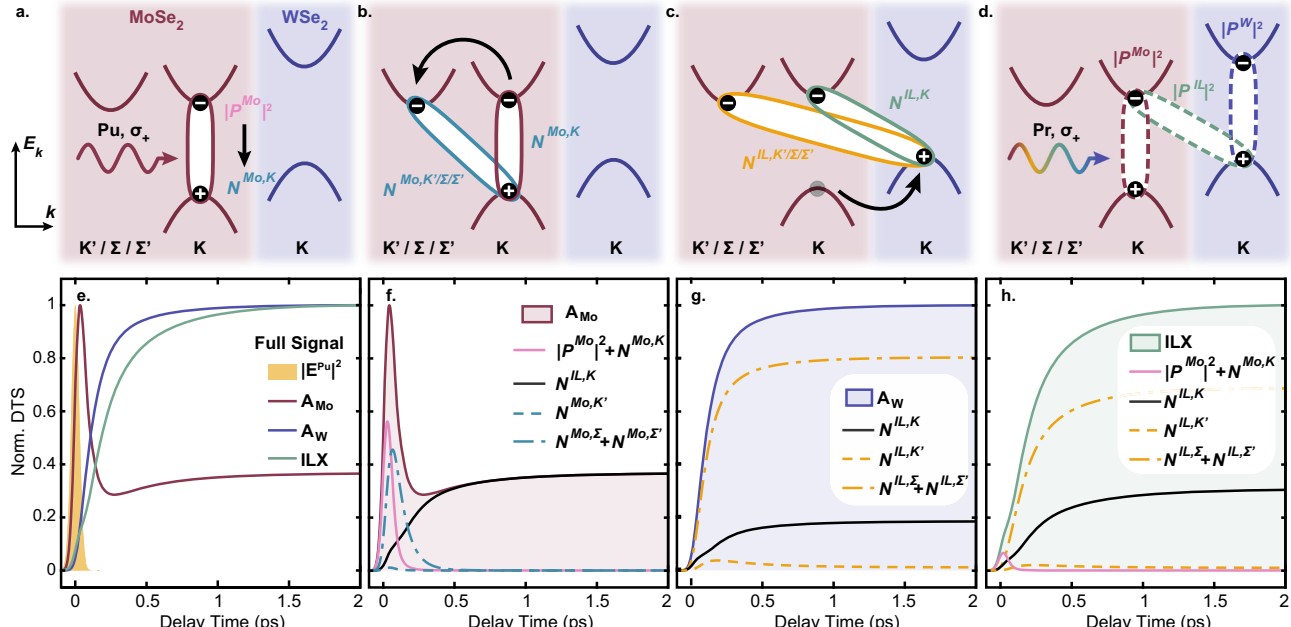

**Fig. 3 | Sketch of inter- and intra- layer scattering processes and calculated DTS.** a–d Cartoon diagram of interlayer charge transfer and interlayer exciton formation dynamics. MoSe$_2$ (WSe$_2$) layer and cartoon bands shown in red (purple). The excitons are represented by full ellipses. **a** The A$_{Mo}$ exciton is optically pumped initially generating a coherent polarization, $P^{Mo}$, and finally the incoherent $K − K$ exciton, $N^{Mo,K}$. **b** Subsequent intervalley electron scattering by phonons leads to the formation of intervalley excitonic populations, $N^{Mo,K'/\Sigma/\Sigma'}$. **c** Phonon-assisted hole tunneling to the WSe$_2$ leads to the formation of momentum direct and indirect ILX populations, $N^{IL,K}$ and $N^{IL,K'/\Sigma/\Sigma'}$. Note that the $K'/\Sigma/\Sigma'$ valleys are not equivalent. **d** The DTS are calculated for the three probed excitonic transitions (dashed ellipses) highlighted in Fig. 2. **e** Calculated DTS for the A$_{Mo}$, A$_W$ and ILX excitons shown in red, purple, and green, respectively, along with the normalized pump pulse in shaded yellow. All the traces are normalized to their maximum. **f–h** Individual intralayer and interlayer excitonic bleaching contributions to the DTS in (**e**).

## Theoretical model

To understand the observed delayed rise in the ILX signal compared to the IHT process, we developed a microscopic model[45,46] for the exciton dynamics in a perfectly aligned (i.e. 0° twist angle) MoSe$_2$/WSe$_2$ HS. A complete picture of the momentum-resolved formation and decay dynamics of excitons is achieved by solving the excitonic Bloch equations for the coherent excitonic polarizations, $P$, and the incoherent excitonic populations, $N$[47]. In our model, we include one optically excited excitonic polarization, $P^{Mo}$, which decays into incoherent exciton populations, $N$, by exciton-phonon scattering processes[48]. The incoherent intralayer population contributions considered in our model, $N^{Mo/W,\xi_e}$, are denoted by a superscript $\xi_e$ corresponding to the valley location of the electron where $\xi_e = K/K'/\Sigma/\Sigma'$ (holes are always located at the $K$ valley of the MoSe$_2$ layer). The considered ILX population is similarly denoted as $N^{IL,\xi_e}$ (hole always at the $K$ valley of the WSe$_2$ layer).

In Fig. 3a–d we sketch the sequence of excitation and scattering processes leading to the formation of the incoherent excitonic populations included in our model. After the optical excitation (Fig. 3a), intraband phonon scattering leads to a decay of exciton polarization, $P^{Mo}$, in expense of the nearly instantaneous formation of an intravalley incoherent exciton population at the $K$ valley of MoSe$_2$ layer, $N^{Mo,K}$. Subsequent intervalley phonon-assisted electron scattering depletes the population $N^{Mo,K}$, leading to the formation of intervalley excitons within the MoSe$_2$, $N^{Mo,K'/\Sigma/\Sigma'}$ (Fig. 3b). In the model, we include electron scattering to the $\Sigma/\Sigma'$ valleys which, in this particular HS, are energetically close to the $K/K'$ valleys because of the strong interlayer hybridization (see Supplementary Note 1.4). Finally, interlayer phonon-assisted hole tunneling to the $K$ valley of WSe$_2$ results in the formation of momentum direct ILX populations, $N^{IL,K}$, and momentum indirect ILX populations, $N^{IL,K'/\Sigma/\Sigma'}$, with electrons located at the $K$ or $K'/\Sigma/\Sigma'$ valleys in the MoSe$_2$ layer and holes located at the $K$ valley in the WSe$_2$ layer (Fig. 3c).

The differential transmission signals (DTS) (Eq. (2) in the Methods section) calculated from these dynamics (Fig. 3a–c) are depicted in Fig. 3d. Different probe energies of A$_{Mo}$, (Eq. (6)), A$_W$ (Eq. (7)), and the momentum-direct ILX (Eq. (8)) are considered. The DTS temporal signatures emerge due to Pauli-blocking processes: Electrons and/or holes forming coherent excitonic polarizations and incoherent excitonic populations ($P^{Mo}$, $N^{Mo,\xi_e}$ and $N^{IL,\xi_e}$) reduce the absorption of the corresponding optical transition and lead to a positive DTS signal. Excitonic effects are manifest in the Pauli-blocking weights (Eqs. (2), (3) in the "Methods" section). These weights are momentum-dependent such that excitonic populations can still contribute to the DTS even if they lie outside the light cone. In this way, excitons encode information on the Pauli blocking of the electron and hole momentum distribution and define the DTS. The calculated DTS timetraces are reported in Fig. 3e together with the bleaching contributions of the individual excitonic populations, Fig. 3f–h. The DTS calculations qualitatively reproduce the experimental dynamics of intralayer excitons and the ILX (Fig. 2c), in particular their different formation timescales. We discuss now the signals at A$_{Mo}$ (i), A$_W$ (ii) and ILX (iii).

(i) The DTS of A$_{Mo}$ exhibits a sharp rise followed by a fast decay and a second delayed rise component (Fig. 3f). The sharp initial peak is mainly due to Pauli-blocking by the instantaneous coherent exciton polarization photoexcited at the $K$ valley (pink line, Fig. 3f) and by the Pauli-blocking weighted incoherent excitonic populations formed in the MoSe$_2$ layer following phonon-mediated dephasing (blue lines, Fig. 3). The second delayed rise component originates from the Pauli-blocking contribution of ILX populations (black line, Fig. 3f) following the decay of intralayer exciton populations via phonon-assisted hole tunneling. This behavior of MoSe$_2$ is also present in the experimental transient optical response of the HS (Fig. 2). A comparison of the DTS signals, both measured and calculated, for the A$_{Mo}$ exciton of an isolated ML and the HS (Supplementary Fig. S16) confirms that the

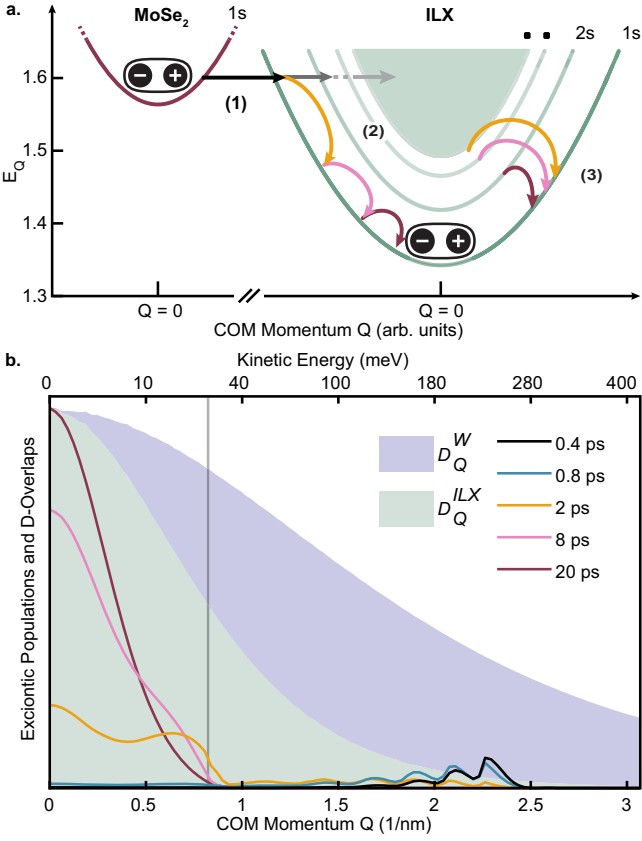

**Fig. 4 | Hot ILX Relaxation. a** Sketch of optically bright ILX formation process following thermal relaxation of hot, momentum dark ILX. The formation process occurs in three steps: (1) After resonant excitation of the $A_{Mo}$ transition, the MoSe₂ layer is populated by incoherent populations, $N_{Q,1s}^{Mo,K}$, (left red parabola) and subsequent phonon-assisted IL hole tunneling leads to the formation of ILX incoherent populations, $N_{Q,\mu}^{IL,K}$, with finite momentum **Q** and also high quantum number $\mu$ (right green parabolas) (see Eq. (1) and Eq. (4)). (2) and (3) The distribution of hot ILX lose energy and momentum by scattering on ultrafast timescales with high energy optical phonons and with low energy acoustic phonons on longer timescales.
**b** Snapshots of the normalized momentum-dependent excitonic populations of ILX at different delays overlaid on the Pauli-blocking weights $D_Q^W$ of the $A_W$ (solid purple), with $i = W, K$, $i' = IL, K$, (see Eq. (3)) and $D_Q^{IL}$ of the ILX (solid green), with $i = IL, K$, $i' = IL, K$ (see Eq. (3)). The distribution in momentum space reflects the different localization of the excitonic wave functions and, thus, the different binding energies of the $A_W$ (215 meV) and of the ILX (91 meV). The binding energy of ILX is the offset between the lowest $1s$ state and the onset of the continuum at $Q=0$. Following IHT, the population distribution of $K − K$ valley ILX, $N_{Q,1s}^{IL,K}$, (lines) is peaked around finite Q values between $2 − 2.5 \, nm^{-1}$, overlapping strongly with $A_W$ Pauli-blocking weight. The overlap of the $N_{Q,1s}^{IL,K}$ with the ILX Pauli-blocking weight increases as they shift to lower energy and momentum via scattering processes described in (**a**). Below the energy threshold of 30 meV (gray line) the scattering is mediated only by acoustic phonons as scattering by optical phonons is energetically forbidden.

presence of ILX populations strongly affects the DTS dynamics of the intralayer $A_{Mo}$ transition.

(ii) The DTS of $A_W$ displays a longer formation time than for $A_{Mo}$ as the signal is not influenced by the excitonic intralayer populations in MoSe₂ (Fig. 3g). Here, only the ILX populations arising from IHT contribute to the Pauli-blocking of the $A_W$ signal.

(iii) The DTS of ILX shows similar behavior as the $A_W$ but with a significantly longer risetime (Fig. 3h). We attribute this longer formation time to the relaxation dynamics of hot ILX populations, represented schematically in Fig. 4a. The difference of the risetimes of the DTS at $A_W$ and at the ILX occurs because hot ILX populations unequally

contribute to the Pauli-blocking of $A_W$ and ILX: Since the binding energies of the ILX (≈100 meV) are lower than the difference between the transition energies of $A_{Mo}$ and ILX (≈300 meV), phonon-mediated hole transfer creates hot ILX populations, $N^{IL,\xi}$ (Fig. 4a). Figure 4 shows how these populations scatter to lower-energy momenta and bound states via optical and acoustic phonons until most of the interlayer populations are in the 1s state. In the following time period, the hot ILX slowly thermalize into a Boltzmann distribution. Figure 4b shows the calculated momentum-dependent ILX populations of the 1s state at different times. Following IHT, the ILX population is initially peaked at higher momenta, after which phonon-assisted scattering processes gradually confine the ILX populations to lower momenta.

This behavior along with the Pauli-blocking weights of the ILX and $A_W$ transitions (green and purple shaded areas, respectively, in Fig. 4b) translates into the DTS and determines the different DTS dynamics. Therefore, the physical origin of the relative delay in the DTS of $A_W$ and ILX can be explained as the following. The larger binding energy of the $A_W$ exciton compared to the ILX leads to a Pauli-blocking weight displaying a broad momentum distribution compared to the ILX Pauli-blocking weight, as shown in Fig. 4b (see Supplementary Note 1.6). This results in the increased sensitivity of the intralayer $A_W$ transition to the overall excitonic population, whereas the probed ILX transition is mainly sensitive to the low-momenta states. Consequently, only the cold and lowest-energy bound ILX populations populated at later delay times significantly contribute to the DTS signal of the ILX transition in Fig. 3h, while the delayed risetime of the ILX compared to the $A_W$ is related to the longer timescale of thermalization of the hot ILX populations. By directly comparing the DTS signals of the probed $A_W$ and ILX transitions, we are able to trace the microscopic mechanisms of the ILX formation and relaxation processes.

All the theoretical results shown so far hold for a HS characterized by a 0° twist angle. Small and finite twist angle effects are not taken into account in our model because we do not expect small twist angles to greatly influence the ILX dynamics. Because of the fermionic character of electron and hole forming the exciton as a quasi-boson, finite center-of-mass (COM) excitons lead to bleaching of the bright excitonic transitions via Pauli blocking and result in DTS dynamics that are only weakly dependent on the twist angle at the relatively small angles deviating from perfectly aligned and anti-aligned stacking[11]. This assumption is supported by transient optical experiments which have shown the timescale of ICT processes (and consequently hot ILX formation) to be weakly dependent on the angular alignment and thus the momentum mismatch between the two layers[49,50].

## Discussion
We have used TA spectroscopy with a combination of high sensitivity and broad spectral coverage to directly track the formation dynamics of ILX in a TMD HS. We find that ILX PB signal rises on a picosecond timescale, significantly slower than the build-up dynamics of intralayer exciton PB signals in general and of IHT specifically. Microscopic calculations reproduce the experimental transient signals and explain the formation timescale in terms of different contributions. The hundreds of femtoseconds build-up time of the WSe₂ PB signal is mainly related to interlayer scattering of hot holes and therefore represents a rather direct estimation of the hot carrier injection process between the TMD layers of the HS. The slower rise time of the ILX signal is the result of the combination of two scattering processes: (1) phonon-mediated exciton cascade process from unbound and/or higher energy excitonic s states to the ground state and (2) intra-exciton energy and momentum relaxation of hot ILX populations. Our simulations also demonstrate that the dynamics of optically bright intra- and inter- layer excitons are influenced by optically dark momentum-indirect excitons.

Besides its fundamental interest, the delayed ILX formation observed and discussed here could explain a long-standing puzzle in optoelectronic devices based on TMD HS, namely the observation of

efficient generation of photocurrent despite the large binding energy of the ILX[51–55]. During their several-hundred femtosecond thermalization process, in fact, the hot ILX have an opportunity to dissociate and form free charge carriers. We foresee that our combined theoretical-experimental approach can be extended to study in real-time exciton formation process in other systems, such as hybrid organic/TMD HS[56] and mixed dimensional van der Waals HS[57].

## Methods

### Sample preparation
Large-area MoSe₂/WSe₂ HS are prepared using a modified gold tape exfoliation method[40] resulting in mm-scale ML flakes deposited onto 200 μm thick transparent SiO₂ substrates. TMD HS with varying twist angles are prepared and characterized using polarization-resolved second harmonic generation (Supplementary Fig. 1). We have mainly reported the static and time-domain measurements on a near aligned HS with a 4° twist angle. Similar measurements were performed on a near anti-aligned HS characterized by 57° twist angle (Supplementary Fig. 18). The HS are prepared with unoverlapped regions where the individual ML can be accessed for control measurements.

### Transient absorption spectroscopy
Time-resolved and static optical measurements are performed at 77 K. TA spectroscopy measurements are performed in a transmission geometry (Supplementary Fig. 2b) with pump and probe beam diameters of 200 μm and 100 μm at the sample surface, respectively. The relatively large sample region accessed in these TA experiments represents a sort of ensemble measurement where sample heterogeneity contributes to the inhomogeneous broadening, as seen previously on similar samples[6]. The delay dependent ΔT/T map and spectra in Fig. 2a, b are measured by dispersing the transmitted broadband probe and acquiring it with a fast silicon spectrometer (Entwicklungsbuero EB Stresing) working at 1 kHz laser repetition rate. The probe pulse is a WLC generated by focusing the output of a homemade NIR Optical Parametric Amplifier (OPA) centered at 1 eV into a YAG plate; the WLC spectrum extends from 1.2 to 2.3 eV, covering A/B intralayer and ILX of the HS (Supplementary Fig. 2a). The NIR OPA is seeded with a regeneratively amplified Ti-sapphire laser (Coherent, Libra) emitting 100-fs pulses at 1.55 eV at a repetition rate of 1 kHz. The pump pulse is generated by a narrowband OPA tuned to the A$_{Mo}$ excitonic resonance ($\hbar\omega$ = 1.59 eV) and modulated at half the repetition rate of the laser. The exciton density of $\approx 3.6 \times 10^{11}$cm$^{-2}$ is calculated using the pump fluence of 2.8 μJ cm$^{-1}$, the pump spectrum and the linear absorption spectrum of the TMD HS at 77 K (Supplementary Fig. 2). The temporal dynamics of the excitonic resonances (Fig. 2c) are measured by a second TA optical setup based on a Yb:KGW regenerative amplifier (Pharos, Light Conversion) providing 200-fs pulses at 1.2 eV and at a higher repetition rate of 100 kHz. The pump pulse is generated by the second harmonic of a near-IR OPA[58] tuned to the same energy and fluence conditions as experiments performed on the lower repetition rate TA setup. The WLC is generated by focusing the fundamental of the laser into a YAG crystal. The transmitted probe beam is then sent to a monochromator and a photodiode for lock-in detection. The pump is modulated by a Pockels cell to 50 kHz allowing to reach higher signal-to-noise ratio with respect to the 1 kHz system. The rise times of the TA signals are estimated by fitting the time traces with a rising exponential convoluted with a Gaussian function with full width at half maximum of the instrument response function (FWHM-IRF) of the respective TA setup. The estimated FWHM-IRF is 140 fs for both TA instruments.

### Theoretical calculations
To obtain the DTS theoretically, we solve the Bloch equations for the excitonic polarizations $P_{\mu,\mathbf{Q}}^{i} = \langle \hat{P}_{\mu,\mathbf{Q}}^{i} \rangle$ up to third order in the electric field as in ref. 45. Here, $\hat{P}_{\mu,\mathbf{Q}}^{i}$ are the excitonic operators[46] carrying excitonic radial quantum number $\mu$, center of mass

momentum **Q** and compound layer and valley index $i = \{l_h, \xi_h, l_e, \xi_e\}$ with electron/hole layer $l_{e/h} = M/W$ for MoSe₂ and WSe₂, electron valley index $\xi_e = K/K'/\Sigma/\Sigma'$ and hole valley index $\xi_h = K$. Performing a linearization in the weak probe pulse limit[59], we are able to separate the pump induced coherent polarizations $P_{\mu,\mathbf{Q}}^{i}$ and incoherent populations

$$N_{\mu,\mathbf{Q}}^{i} = \langle \hat{P}_{\mu,\mathbf{Q}}^{\dagger,i} \hat{P}_{\mu,\mathbf{Q}}^{i} \rangle_c \tag{1}$$

from the probe-induced dynamics. The coupled equations of motion[47] are then solved in the time domain. The hole tunneling process is implemented as in ref. 60. The subscript c in Eq. (1) accounts for the purely correlated (or incoherent) part of the expectation value in the spirit of ref. 47,61. By assuming a Dirac delta-shaped probe pulse, we find an expression for the probe- and pump-induced macroscopic polarization $\mathbf{P}^{i} = \frac{1}{A} \sum_{\mu} \mathbf{d}_{\mu}^{i} P_{\mu,\mathbf{Q}=\mathbf{0}}^{i}$ which couples to Maxwell's equations giving the total electric field and therefore the transmission at the HS as in ref. 62,63. Here, $\mathbf{d}_{\mu}^{i}$ is the excitonic dipole moment and $A$ is the illuminated area of the sample. The DTS is then given by subtracting the transmission $T^t$ of the probe pulse without pump pulse from the transmission $T^{t+p}$ of the probe pulse with pump pulse: $\Delta T(\omega, \tau) = T^{t+p}(\omega, \tau) - T^t(\omega)$[64]. In our case, since we are only interested in timetraces, we neglect Coulomb renormalization and nonlinear broadening[65] and focus solely on the Pauli-blocking induced PB dynamics, so that the DTS signal dependent on the time delay $\tau$ between pump and probe pulses can be expressed as

$$\Delta T_{\mu}^{i}(\tau) \sim \sum_{i',\mathbf{Q},\mu} \left( D_{\mathbf{Q},\mu,\nu}^{e,i,i'} + D_{\mathbf{Q},\mu,\nu}^{h,i,i'} \right) \left( |P_{\nu,\mathbf{Q}}^{i'}(\tau)|^2 \delta_{\mathbf{Q},\mathbf{0}} + N_{\nu,\mathbf{Q}}^{i'}(\tau) \right), \tag{2}$$

where the matrix elements are given by

$$D_{\mathbf{Q},\mu,\nu}^{e/h,i,i'} = \sum_{\mathbf{q}} \mathbf{d}_{\mu}^{i} \cdot \mathbf{d}_{\mathbf{q}}^{cv,i} \varphi_{\mu,\mathbf{q}}^{*,i} \varphi_{\nu,\mathbf{q}+(-\alpha_{i'})/(+\beta_{i'})\mathbf{Q}}^{*,i'} \varphi_{\nu,\mathbf{q}+(-\alpha_{i'})/(+\beta_{i'})\mathbf{Q}}^{i'} \delta_{l_{h/e},l_{h/e}'} \delta_{\xi_{h/e},\xi_{h/e}'}. \tag{3}$$

Here, $\mathbf{d}_{\mu}^{i} = \sum_{\mathbf{q}} \varphi_{\mu,\mathbf{q}}^{i} \mathbf{d}_{\mathbf{q}}^{*,cv,i}$ are the excitonic and $\mathbf{d}_{\mathbf{q}}^{cv,i}$ the electronic dipole moments, $\varphi_{\mu,\mathbf{q}}^{i}$ are the excitonic wave functions obtained by solving the Wannier equation for the respective excitonic configuration $i$ with radial quantum number $\mu$ and relative momentum $\mathbf{q}$. $\alpha_i$ and $\beta_i$ are the ratios of the effective masses[37]. In Eq. (3), the indices $\mu$ and $i$ refer to the optically bright probed transition, whereas $\nu$ and $i'$ reflect the bright as well as dark excitonic populations responsible for the Pauli blocking. These matrix elements can be viewed as convolutions in momentum space of the excitonic wave functions $\varphi_{\mu,\mathbf{q}}$ of the probed transitions $i$ with the excitonic wave functions $\varphi_{\nu,\mathbf{q}}^{i}$ of the pumped populations $i'$ and constitute the crucial part of the explanation regarding the different DTS risetimes of the A$_W$ and ILX transition as described in the Results section. The Wannier equation is solved for a HS using a Coulomb potential for two dielectric slabs in three dielectric environments, $\epsilon_1 = 3.9$ (SiO₂ substrate on the MoSe₂ side), $\epsilon_g = 1$ (assuming a vacuum environment between the layers) and $\epsilon_2 = 1$ (vacuum on the WSe₂ side), as in ref. 66. Our calculation includes both high energy bound exciton interlayer $s$ states and unbound states above the band edge.

In the following, and to be in line with the notation used in the main part of the manuscript, we denote the excitonic populations under consideration as:

$$N^{Mo,\xi_e} = N^{M,K,M,\xi_e}, N^{IL,\xi_e} = N^{W,K,M,\xi_e}, \tag{4}$$

and the excitonic coherent polarizations as:

$$P^{Mo} = P_{1s}^{M,K,M,K}. \tag{5}$$

Therefore, the DTS signal for the probed $A_{Mo}$ transition reads explicitly:

$$\Delta T^{A_{Mo}}(\tau) \sim \sum_{\mathbf{Q}} \left( D^{e+h,Mo,K,Mo,K}_{\mathbf{Q},1s,1s} \left( |P^{Mo}(\tau)|^2 \delta_{\mathbf{Q},\mathbf{0}} + N^{Mo,K}_{\mathbf{Q},1s}(\tau) \right) \right.$$
$$\left. + \sum_{\xi_e \neq K} D^{h,Mo,K,Mo,\xi_e}_{\mathbf{Q},1s,1s} N^{Mo,\xi_e}_{\mathbf{Q},1s}(\tau) + \sum_{\mu} D^{e,Mo,K,IL,K}_{\mathbf{Q},1s,\mu} N^{IL,K}_{\mathbf{Q},\mu}(\tau) \right), \quad (6)$$

for the probed $A_W$ transition we obtain:

$$\Delta T^{A_W}(\tau) \sim \sum_{\mathbf{Q},\mu,\xi_e} D^{h,W,K,IL,\xi_e}_{\mathbf{Q},1s,\mu} N^{IL,\xi_e}_{\mu,\mathbf{Q}}(\tau), \quad (7)$$

and for the probed ILX transition it reads:

$$\Delta T^{ILX}(\tau) \sim \sum_{\mathbf{Q}} \left( D^{e,IL,K,Mo,K}_{\mathbf{Q},1s,1s} \left( |P^{Mo}(\tau)|^2 \delta_{\mathbf{Q},\mathbf{0}} + N^{Mo,K}_{1s,\mathbf{Q}}(\tau) \right) \right.$$
$$\left. + \sum_{\mu} D^{e+h,IL,K,IL,K}_{\mathbf{Q},1s,\mu} N^{IL,K}_{\mu,\mathbf{Q}}(\tau) + \sum_{\mu,\xi_e \neq K} D^{h,IL,K,IL,\xi_e}_{\mathbf{Q},1s,\mu} N^{IL,\xi_e}_{\mu,\mathbf{Q}}(\tau) \right). \quad (8)$$

Equation (6) displays the contribution of the probed $A_{Mo}$ transition (red ellipse in Fig. 3d) with Pauli blocking due to electron and hole of the pumped momentum-direct excitonic polarizations $P^{Mo}$ and populations $N^{Mo,K}$ (first two terms) as well as due to the blocking of the hole of momentum-indirect excitonic populations $N^{Mo,K'/\Sigma/\Sigma'}$ (third term) and blocking of the electron of momentum-direct interlayer populations $N^{IL,K}$ (last term). Equation (7) describes the probed $A_W$ transition (purple ellipse in Fig. 3d) with Pauli blocking contributions due to holes of all four possible interlayer populations $N^{IL,\xi_e}$. Equation (8) shows the probed ILX transition (green ellipse in Fig. 3d), where the first two terms account for Pauli blocking due to the electron of the pumped momentum-direct excitonic polarizations $P^{Mo}$ and populations $N^{Mo,K}$, the third term accounts for the blocking of electron and hole of the momentum-direct interlayer population $N^{IL,K}$, whereas the last term shows the contribution of the PB by the hole of the other momentum-indirect interlayer populations $N^{IL,K'/\Sigma/\Sigma'}$. All Pauli blocking terms feature distinctive momentum- and excitonic quantum number-dependent Pauli-blocking weights $D^{e/h,i,i'}_{\mathbf{Q},\mu,\nu}$, which are given in Eq. (3).

## Data availability

The processed spectroscopic data of the manuscript are available at https://doi.org/10.6084/m9.figshare.24428992. Supplementary information data will be made available by request to the corresponding authors.

## Code availability

Code pertaining to the theoretical model in this work will be made available by request to the corresponding authors.

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

## Acknowledgements

We are grateful to C. J. Sayers for helpful discussions and acknowledge F. Morabito, C. Trovatello, A. Genco, S. Sardar and C. D'Andrea for experimental contributions. We would like to thank the IT and data services members of Zuse Institute Berlin for providing the computing infrastructure. S.D.C. acknowledge financial support from MIUR through the PRIN 2017 Programme (Prot. 20172H2SC4). G.C. acknowledge support by the European Union Horizon 2020 Programme under Grant Agreement 881603 Graphene Core 3. X.Y.Z. acknowledges support for sample fabrication by the Materials Science and Engineering Research Center (MRSEC) through NSF grant DMR-2011738. F.S. and A.V. acknowledge support by the European Research Council (ERC) under the European Union's Horizon 2020 research and innovation programme (grant agreement No. 816313). We acknowledge financial support from the Deutsche Forschungsgemeinschaft (DFG) through SFB 951 Project No. 182087777 (M.K., M.S., and A.K) and Project KN 427/11-2 (H.M. and A.K.) Project No. 420760124.

## Author contributions

V.R.P. and S.D.C. devised the experimental work. V.R.P., O.D., and A.V. performed the experimental work; V.R.P. and O.D. performed data analysis. H.M., M.K., and B.K. performed the theoretical work. Q.L. prepared and characterized the TMD HS samples. All authors contributed to the interpretation of the results and preparation of the manuscript.

## Competing interests

The authors declare no competing interests.
