## [Peer Review File · Nature Communications]

Reviewers' Comments:

Reviewer #1:

Remarks to the Author:

Policht et al. report on new insights with regard to the formation and thermalization dynamics of interlayer excitons (ILX) in TMD heterostructures. As the authors introduce, so far, only a few experimental studies have had a direct access to the ILX dynamics. Based on an advanced ultrafast optical TA setup, Policht et al. provide significant new experimental insights on this topic: With the identification of a spectral signature that is directly related to ILX ($\sim 1.4\text{eV}$), it is possible to directly evaluate its dynamics in a femtosecond time-resolved experiment. The experimental signatures for the different dynamics of the TA signal from the ILX (800 fs) and the photo-bleaching from the intralayer excitons (200 fs) are unambiguous. Based on microscopic model calculations, the authors then explain the different dynamics with a mechanism that first considers IHT leading to the formation of ILX. Subsequently, the ILX thermalize into the lightcone and can be detected in the TA experiment.

Because of the distinct access to the femtosecond thermalization dynamics of ILX, I think the manuscript is highly relevant for all research efforts on the ultrafast dynamics of moiré heterostructures. Hence, it is suitable for publication in Nature Communications. However, I would appreciate very much if the authors could address the following comments and questions.

- On page 4, the authors state that optically-bright ILX are not formed on the same timescale as the ICT occurs. When inspecting Fig.4a, I can understand this statement because, after ICT, subsequent phonon-mediated energy relaxation processes have to transfer the excitonic occupation into the lightcone of the ILX. While this is true for the model that calculates with a perfectly aligned heterostructure (K_W valley on K_Mo valley), in how far does this argument hold for the 4° twisted sample in experiment (K_W and K_Mo valleys are offset in momentum). In think the discussion of this difference between experiment and theory could make the manuscript even stronger.
- On page 5 and Fig. 3b/c, the authors discuss the formation of intervalley Σ excitons in MoSe₂. In the MoSe₂/WSe₂ heterostructure, I would expect that the electron contribution to the Σ excitons is delocalized over both layers, i.e., that the exciton is of hybrid character. This should lead to a redshift of its exciton energy. Can the authors comment on this? Is the hybrid character of the Σ excitons included in the microscopic model?
- Depending on the energy alignment of the bright K exciton and the Σ exciton, a different steady state between both excitons will be established on the sub-1 ps timescale because of phonon emission and absorption processes mediating the transfer of excitonic occupation between both. How does this steady state change the ICT?
- In the MoSe₂/WSe₂ heterostructure, the valence bands at the Γ are hybridized and energetically favorable over the K valley valence band of MoSe₂. An increasing amount of experimental and theoretical literature suggests that charge transfer occurs in a step-wise process via such hybridized intermediate states. Can the authors comment on this alternative IHT mechanism? If I understand correctly, it is not included in the microscopic model?
- In how far should moiré effects and renormalized excitonic bands be considered in the ILX thermalization process on the small twist-angle sample?

Minor comments:

- The labelling of a/b/c of Fig. 1 got mixed up on page 3.
- Can the authors estimate the exciton density? In the bottom of page 4, the authors write that they are well below the Mott threshold. However, in the top of page 4, they comment on an instantaneous renormalization of the exciton resonance. While nice data is provided in Fig. S8, this point could be clarified in the main text.
- On page 6 and Fig.4a, why do the authors argue with the binding energy of the ILX (and not directly E_Q)? If so, I think it would be very helpful to indicate the binding energies in Fig. 4a.

Reviewer #2:

Remarks to the Author:

V. R. Policht et al. report on the formation time of interlayer excitons (ILX) in MoSe₂/WSe₂ heterostructures upon direct excitation of MoSe₂ A exciton which is notably shorter than the apparent charge transfer time. Through transient absorption, they extract the ILX formation time by probing the ILX absorption and the charge transfer time by probing the WSe₂ absorption. This is a unique result in the field, and, to my knowledge, no one has reported on the difference in charge transfer and formation of the ILX population. Understanding this dynamic behavior is essential in building optoelectronic devices which harness the ILX. However, to be a properly supported and methodologically sound report a few technical considerations must be addressed. Bigger concerns:

1. The major conclusions of the paper depend on the absorption of the ILX being less sensitive to hot charge carriers than the absorption of WSe₂. The main rationale behind this is that the binding energy is smaller for ILXs, leading to a larger radius and smaller momentum spread. They conclude that since the thermalized WSe₂ exciton has a larger momentum spread, the WSe₂ absorption is more sensitive to hot carriers. However, this explanation seems to ignore the fact that the incoming photon probing absorption has its limited to a range in momentum space much smaller ($\sim 0.01/\text{nm}$) than the thermalized spread of both ILX and WSe₂ exciton. Further explanation of the difference in between WSe₂/ILX absorption in the context of optically accessible exciton states is needed.

2. Is the long formation time unique to hole-transfer in WSe₂/MoSe₂ bilayers? A small additional measurement that the authors could perform (with existing samples) is exciting the WSe₂ and looking at electron charge transfer which may show different formation times due to scatter with the other conduction band minimum. Even though MoSe₂ is a smaller bandgap, it has very little absorption at the WSe₂ resonance.

Medium concerns:

3. On page 3, paragraph 2, the authors attribute the broad ILX emission peak (in Figure 1c) to the contribution of both momentum-direct and momentum-indirect transitions. If this were a significant contribution, we would expect the absorption to be much narrower, which does not appear to be the case in Figure 2b. It seems more likely that they are probing sample inhomogeneity.

4. In Figure 1c, the power density and wavelength used to generate the photoluminescence are not reported.

5. On page 4, paragraph 2, the authors attribute the blueshift from the absorption to the emission to the energy difference between momentum-direct and -indirect transitions. While this is a possible explanation, the sample inhomogeneity can explain the Stark shift more simply. Exciton diffusion, particularly of hot excitons, allows the excitons to find the energetic minima in the potential landscape whereas absorption will probe the states available and thus on average will probe a higher energy.

Minor concerns:

6. The authors refer to the indirect transition as one between K- Σ as in the conduction band minima is at Σ . Given the plethora of notations of this point of the Brillouin zone (which sometimes called Q/ Λ in papers such as Hanbicki et al. ACS Nano, 12(5) 2018 and Rivera et al. Nature Nanotech 13, 2018), it would be useful to define its position in momentum space as you introduce it.

7. The dashed/dot-dashed lines in Fig. 3f,g, and h are visually hard to distinguish from one another.

Reviewer #3:

Remarks to the Author:

In this manuscript, the authors studied the interlayer exciton formation dynamic process. They found that the interlayer excitons formation timescale is nearly an order of magnitude (~ 1 ps) longer than the interlayer charge transfer time (~ 100 fs). They attributed the relative delay to an interplay between a phonon-assisted interlayer exciton cascade and subsequent cooling processes. The results are important for exciton physics. Although this work has been carried out with care, some issues still exist.

The interlayer exciton formation dynamics are really difficult to access directly due to the weak oscillator strength and the ultrafast interlayer charge transfer process. However, in this manuscript, the signal strength $\Delta T/T$ is very very weak in the range of interlayer exciton. The

signal needs to be amplified a hundred times to see the outline. Moreover, photobleaching signal peaks generally increase on the picosecond scale (Nature Photonics 2020,14, 171-176; 2021,15, 238-244). Further proof is needed that this is not caused by noise signal. The relaxation process of hot interlayer exciton should take different times at different temperatures. As the authors cite the literature 24 (Nano Lett. 2021, 21, 2165-2173), the authors can consider other temperatures to confirm further the reliability of this signal, not just 77 K.

Reviewer Reply: Time-domain observation of interlayer exciton formation and thermalization in a MoSe₂/WSe₂ heterostructure

REVIEWER COMMENTS

Reviewer #1 (Remarks to the Author):

Policht et al. report on new insights with regard to the formation and thermalization dynamics of interlayer excitons (ILX) in TMD heterostructures. As the authors introduce, so far, only a few experimental studies have had a direct access to the ILX dynamics. Based on an advanced ultrafast optical TA setup, Policht et al. provide significant new experimental insights on this topic: With the identification of a spectral signature that is directly related to ILX (1.4eV), it is possible to directly evaluate its dynamics in a femtosecond time-resolved experiment. The experimental signatures for the different dynamics of the TA signal from the ILX (800 fs) and the photo-bleaching from the intralayer excitons (200 fs) are unambiguous. Based on microscopic model calculations, the authors then explain the different dynamics with a mechanism that first considers IHT leading to the formation of ILX. Subsequently, the ILX thermalize into the lightcone and can be detected in the TA experiment. Because of the distinct access to the femtosecond thermalization dynamics of ILX, I think the manuscript is highly relevant for all research efforts on the ultrafast dynamics of moiré heterostructures. Hence, it is suitable for publication in Nature Communications. However, I would appreciate very much if the authors could address the following comments and questions.

We thank the Reviewer for judging our results “highly relevant,” and for recommending publication of the paper in Nature Communications. In the following we fully address the Reviewer’s concerns.

- On page 4, the authors state that optically-bright ILX are not formed on the same timescale as the ICT occurs. When inspecting Fig.4a, I can understand this statement because, after ICT, subsequent phonon-mediated energy relaxation processes have to transfer the excitonic occupation into the lightcone of the ILX. While this is true for the model that calculates with a perfectly aligned heterostructure (K_W valley on K_{Mo} valley), in how far does this argument hold for the 4° twisted sample in experiment (K_W and K_{Mo} valleys are offset in momentum). I think the discussion of this difference between experiment and theory could make the manuscript even stronger.

We thank the reviewer for this comment and for the suggestion to point out possible differences between the experimental measurements performed on small degree twist angle and the simulations performed on perfectly aligned HS (i.e. 0°). We agree with the reviewer that only bright excitons, i.e. excitons with zero center of mass momentum (COM), can directly couple with visible light. For ILX, this happens only when phonon-mediated scattering processes transfer excitonic occupation into the light cone of ILX, after the creation of hot (and therefore finite COM) ILX population. However, in optical pump-probe measurements, optically bright excitonic transitions can be perturbed or “bleached” by Pauli blocking processes because they share the electron or the hole part of dark (i.e. finite COM) excitonic transitions (see equations 6-8). In this sense, the dynamics of bright excitons are indirectly determined by dark excitons even if the electron or the hole part of their excitonic wavefunctions lies outside the light cone. Our HS is characterized by a small interlayer twist angle which gives rise to a small momentum offset between K_{Mo} and K_W valleys and a finite COM for the $K - K$ ILX. Given what has been said before (i.e. the fact that bright and finite COM dark excitons can share the electron or the hole part of their excitonic wavefunctions), we do not expect a significant difference in the dynamics of either the intra- or inter-layer excitons in our experimentally measured small twist angle HS compared to the modeled 0° HS. This assumption is also supported by previous optical pump-probe experiments [1] which have shown that the timescale of the ICT process (which determines the hot ILX formation) is independent of the angular alignment and thus the momentum mismatch between the two layers, when studied with a temporal resolution comparable to our instrumental response function (i.e. 100 fs). Also, recent time-resolved optical measurements, performed with improved temporal resolution (i.e. below 20 fs), have shown that the interlayer charge transfer time can weakly increase with respect to zero alignment geometry but only for large values of the twist angle [2]. ACTION TAKEN: In the new version of the manuscript, we replace the sentence “With this approach, PB signatures by electrons and/or holes are fully taken into account in the excitonic model” (pg. 5) with the following sentences:

We stress that in our theoretical model, ILX with finite momenta Q can contribute via Pauli blocking process to the DTS signals calculated at the energies of bright (i.e. momentum direct) excitonic transitions, even if they lie outside the light cone (see eq. 2-3 in Methods section). This is due to the fact that bright excitons share their electron or hole part with finite momenta excitonic states.

Regarding the discussion of the possible difference between experiment and theory for HS at finite twist angles, we add the following sentence at the end of the manuscript on pg. 7:

All the theoretical results shown so far hold for an HS characterized by a zero degree twist angle. Small and finite twist angle effects are not taken into account in our model due to the increased computational complexity and because we do not expect small twist angles to greatly influence the ILX dynamics. Because of their fermionic character, finite COM excitons lead to bleaching of the bright excitonic transitions via Pauli blocking and result in PB dynamics that are only weakly dependent on the twist angle. This assumption is supported by transient optical experiments which have shown the timescale of ICT processes (and consequently hot ILX formation) to be weakly dependent on the angular alignment and thus the momentum mismatch between the two layers [1, 2].

- On page 5 and Fig. 3b/c, the authors discuss the formation of intervalley Σ excitons in MoSe_2 . In the $\text{MoSe}_2/\text{WSe}_2$ heterostructure, I would expect that the electron contribution to the Σ excitons is delocalized over

both layers, i.e., that the exciton is of hybrid character. This should lead to a redshift of its exciton energy. Can the authors comment on this? Is the hybrid character of the Σ excitons included in the microscopic model?

We thank the reviewer for this question. We stress that our theory treats the tunnel interaction perturbatively as in ref. [3]. This approach is different from the one adopted in ref. [4, 5] where the exciton hybridization is explicitly included in the theoretical model and the exciton Hamiltonian is diagonalized on the basis of hybrid excitons. Our microscopic model does not explicitly include hybridization of the exciton wavefunctions but hybridization is indirectly included in our calculations. In particular, the energetic position of the Σ valley of the MoSe₂ layer is treated as a parameter and is tuned to reproduce the measured pump-probe signal. Our results find that the Σ valley is redshifted from its un-hybridized energy by about 160 meV. This energy shift agrees quite well with the interlayer hopping energy at the Σ valleys reported in Ref [4].

ACTION TAKEN: We add the following sentence regarding the hybrid character of the exciton in Supplementary Note 1.4

In MoSe₂/WSe₂ HS the Σ valleys experience a red shift due to the strong hybridization between the layers [4]. The tunnel interaction in Eq. (S8) is treated perturbatively in our theoretical model such that band hybridization is included indirectly. We treat the energy position of the Σ valleys in the MoSe₂ layer as a parameter which is tuned to reproduce the measured pump-probe signal. The resulting energy of the Σ valleys is effectively red-shifted by about 160 meV with respect to the energy position without interlayer hybridization in the ML. This energy shift agrees quite well with the interlayer hopping energy at the Σ valleys reported in Ref. [4].

- Depending on the energy alignment of the bright K exciton and the Σ exciton, a different steady state between both excitons will be established on the sub-1 ps timescale because of phonon emission and absorption processes mediating the transfer of excitonic occupation between both. How does this steady state change the ICT?

We thank the reviewer for the comment. The ICT rates proceeding from the K - K and K - Σ intralayer excitons take place on similar timescales of 54 fs and 62 fs, respectively. More precisely, the K - K and K - Σ ICT processes reflect the hole tunneling from the K - K and K - Σ intralayer excitons to K point in the conduction band, forming the ILX populations (i.e. the green and yellow ellipses in Fig. 3c of the main manuscript). The timescales of these ICT processes are too rapid to form a steady state and do not appear directly in the calculated DTS traces due to the interplay with other excitonic species. To address the question about the contribution of the scattering processes involving the Σ valleys on the ICT timescale, we calculated the total ILX population dynamics in three cases presented in Fig. R1: (1) as presented in the main text with the Σ valley and higher lying Rydberg states (green); (2) with higher lying Rydberg states and no Σ valley (blue, dashed); (3) only $1s$ states with the Σ valley (yellow, dotted). We find that the scattering processes involving high-energy excitonic Rydberg states (blue) and Σ valleys (yellow) each contribute to speed up the ILX population compared to the case where only K - K valleys are taken into account. However, the formation rate of ILX population is much faster with the inclusion of higher energy s states of the Rydberg series (Fig. R1 blue) compared to the inclusion of the Σ valley (Fig. R1 yellow). This suggests a weaker, but non negligible, contribution from the Σ valley states to the ICT process.

ACTION TAKEN: We have added Figure R1 to the Supplementary Information and a discussion in Supplementary Note 1.5 titled "Σ valleys and high Rydberg *s* states contribution to ILX population dynamics:"

We consider the contribution of different scattering processes to the formation dynamics of the ILX population. In Supplementary Fig. S5, we display the total ILX population dynamics calculated for three cases: (1) as presented in the main text with inclusion of the Σ valley and higher lying Rydberg states (green); (2) with higher lying Rydberg states and no Σ valley (blue, dashed); (3) only *1s* states with the Σ valley (yellow, dotted). These calculations show that scattering processes involving high-energy excitonic Rydberg states (blue) and Σ valleys (yellow) each contribute to speed up the ILX population compared to the case where only *K-K* valleys are taken into account. However, the formation rate of ILX population is much faster with the inclusion of higher energy *s* states of the Rydberg series (Supplementary Fig. S5, blue) compared to the inclusion of the Σ valley (Supplementary Fig. S5, yellow). This suggests a weaker contribution from the Σ valley states to the ICT process and dominant contribution of high-energy-lying Rydberg *s* states.

Fig. R 1 ILX population dynamics. The temporal dynamics of ILX population is calculated including the scattering processes from momentum-direct (*K-K*) *1s* states with different combinations of momentum-indirect (*K-Σ*) excitons and higher-lying interlayer Rydberg excitons. The best agreement with experimental data requires inclusion of both higher-lying Rydberg states and the Σ valley (green). A slower ILX build-up is seen in the conditions where only either the higher-lying Rydberg states (blue, dashed) or the Σ valleys (yellow, dotted) are included.

- In the MoSe₂/WSe₂ HS, the valence bands at the Γ are hybridized and energetically favorable over the K valley valence band of MoSe₂. An increasing amount of experimental and theoretical literature suggests that charge transfer occurs in a step-wise process via such hybridized intermediate states. Can the authors comment on this alternative IHT mechanism? If I understand correctly, it is not included in the microscopic model?

We thank the reviewer for pointing out a possible additional hole scattering process involving electronic states at Γ valley. It has been shown by theoretical calculations [5] that in some HS, such as MoS₂/WS₂, holes at the K valleys undergo a strong tunnelling to the Γ valley as a consequence of the strong hybridization between layers. This leads to an energetic blue-shift of the valence band states at the Γ point such that Γ-K is the lowest lying excitonic state. The band structure of MoSe₂/WSe₂ HS differs from the previous HS. In this HS, the Σ-K exciton is nearly energetically degenerate with *K-K* exciton due to the strong hybridization of the electronic

states at Σ valley, that leads to an energetic red-shift of the conduction band state at the Σ point[4]. In this case, the electrons belonging to K - K MoSe₂ excitons can scatter from the K to the Σ valley more efficiently than the holes can scatter from the K to the Γ valley. Moreover, the deformation potential (which determines the electron-phonon interaction strength and the intervalley scattering rates), is higher for K - Σ electron transition compared to Γ - K hole transition [6]. Furthermore, there is only one Γ valley and six Σ/Σ' valleys within the first Brillouin zone, which further renders the K - Σ scattering channel more effective than the Γ - K channel. For all these reasons, and considering the dominant contribution of high-energy Rydberg excitons over momentum-indirect (K - Σ) excitons, we expect that the inclusion of hole scattering process involving the Γ valley as an intermediate state has a minor effect on the ILX population dynamics [7].

ACTION TAKEN: We have added the following discussion in Supplementary Note 1.5:

Theoretical calculations have shown that both the lower conduction band state at Σ and the higher valence band state at Γ have strong hybrid character of the constituent MLs [4]. The momentum-indirect Γ - K hybrid exciton is almost degenerate with K - K interlayer exciton in MoS₂/WS₂ HS and has been implicated as an intermediate state in charge transfer processes [5] as well as in MoS₂/WSe₂ [2]. In MoSe₂/WSe₂ HS, however, the hybridization effect is stronger for Σ valleys and K - Σ excitons are the energetically lowest states for different interlayer stackings [4]. Because of this exciton energy landscape, the electrons belonging to K - K MoSe₂ excitons are expected to scatter from the K to the Σ valley more efficiently than the holes from K to Γ valley. Furthermore, there is only one Γ valley and six Σ/Σ' valleys within the first Brillouin zone, which further renders the K - Σ scattering channel much more effective than the K - Γ channel. For all these reasons, we expect a minor contribution of the hole scattering process involving the Γ valley as an intermediate state in the formation dynamics of the ILX population [7].

- In how far should moiré effects and renormalized excitonic bands be considered in the ILX thermalization process on the small twist-angle sample?

We thank the reviewer for this comment. Signatures of interlayer moiré excitons are expected to appear in the emission spectrum of high-quality low-twist angle TMD HS embedded in hBN at low temperature (i.e. 4K). PL measurements have shown these moiré IIX signatures to drastically diminish at temperatures well below 77 K (i.e. 19 K) [8]. In our case, the sample heterogeneity (due to lack of encapsulation and intrinsically high density of defects of the monolayers) and the high temperature at which we performed the measurements (i.e. 77K) prevented us from observing clear moiré effects.

ACTION TAKEN: We have added a phrase to the main text on pg. 3 to explicitly state that our measurements are inhomogeneously broadened due to our beam spot diameters:

The pump and probe are focused to spot diameters on the order of 100 μm , such that sample heterogeneity contributes to inhomogeneous broadening of the TA peaks.

Minor comments:

- The labelling of a/b/c of Fig. 1 got mixed up on page 3.

We thank the reviewer for pointing out this error.

ACTION TAKEN: We thank the reviewer for the comment. We have corrected the subfigure references on page 2 and 3, highlighted in red in the revised version of the manuscript.

- Can the authors estimate the exciton density? In the bottom of page 4, the authors write that they are well below the Mott threshold. However, in the top of page 4, they comment on an instantaneous renormalization of the exciton resonance. While nice data is provided in Fig. S8, this point could be clarified in the main text.

We estimate the exciton density to be $3.6 \times 10^{11} \text{ cm}^{-2}$ based on the incident pump fluence and the linear absorption spectrum reported in Fig. 1b.

ACTION TAKEN: We have added the estimation of the exciton density to page 4 of the main text:

The peak we observe at 1.37 eV is not related to this effect because (i) the photoinduced exciton density (i.e. $\approx 3.6 \times 10^{11} \text{ cm}^{-2}$) is well below the Mott threshold...

- On page 6 and Fig.4a, why do the authors argue with the binding energy of the ILX (and not directly E_Q)? If so, I think it would be very helpful to indicate the binding energies in Fig. 4a.

We thank the reviewer for the comment and regret the miscommunication. Both the transition energies (E_Q) and the binding energies of the excitons will impact the observed TA signals. The difference between the energy of A_{Mo} exciton and ILX set the energy of the hot ILX population at early delay times (see Fig. 4a). Conversely, the binding energies of the WSe_2 intralayer exciton and the ILX determine the distribution of their wavefunctions in momentum space (see Fig. 4b), and therefore impact the onset of the observed WSe_2 A and ILX signals in TA. The binding energies for these two excitons are 215 meV and 92 meV and are calculated as the solutions of the Wannier equation for the HS.

ACTION TAKEN: We have explicitly reported the exciton binding energies in the caption of Fig. 4 to more directly highlight the role of exciton binding energy on the observed dynamics:

The distribution in momentum space reflects the different localization of the excitonic wave functions and, thus, the different binding energies of 215 meV (A_W) and 91 meV (ILX). The binding energy of ILX is the offset between the lowest $1s$ state and the onset of the continuum at $Q=0$.

We have additionally modified the following sentence on page 6 to clarify our argument (original in blue, changes in red):

Since the binding energies of the ILX (≈ 100 meV) are lower than the difference between the transition energies of A_{Mo} and ILX (≈ 300 meV), phonon-mediated hole transfer creates hot ILX populations, i.e. populations at high momenta \mathbf{Q} and/or higher energy bound and unbound s states of the Rydberg series (Fig. 4a), which subsequently scatter to lower-energy momenta and bound states via optical and acoustic phonons. Due to the large energy mismatch between the excitonic energies of A_{Mo} and ILX (≈ 300 meV) phonon-mediated hole transfer creates hot ILX populations, i.e. populations at high momenta \mathbf{Q} and higher-lying bound Rydberg states. Unbound ILX above the continuum can also be

generated, because the energy mismatch is larger than the binding energy of the ILX (Fig. 4a). All the ILX populations subsequently scatter to $1s$ ILX ground state via optical and acoustic phonons.

Reviewer #2 (Remarks to the Author):

V. R. Policht et al. report on the formation time of interlayer excitons (ILX) in MoSe₂/WSe₂ heterostructures upon direct excitation of MoSe₂ A exciton which is notably shorter than the apparent charge transfer time. Through transient absorption, they extract the ILX formation time by probing the ILX absorption and the charge transfer time by probing the WSe₂ absorption. This is a unique result in the field, and, to my knowledge, no one has reported on the difference in charge transfer and formation of the ILX population. Understanding this dynamic behavior is essential in building optoelectronic devices which harness the ILX. However, to be a properly supported and methodologically sound report a few technical considerations must be addressed.

We thank the Reviewer for considering our results unique in the field and essential for building optoelectronic devices based on ILX. In the following we fully address the Reviewer's concerns.

Bigger concerns:

1. The major conclusions of the paper depend on the absorption of the ILX being less sensitive to hot charge carriers than the absorption of WSe₂. The main rationale behind this is that the binding energy is smaller for ILXs, leading to a larger radius and smaller momentum spread. They conclude that since the thermalized WSe₂ exciton has a larger momentum spread, the WSe₂ absorption is more sensitive to hot carriers. However, this explanation seems to ignore the fact that the incoming photon probing absorption has its limited to a range in momentum space much smaller (0.01/nm) than the thermalized spread of both ILX and WSe₂ exciton. Further explanation of the difference in between WSe₂/ILX absorption in the context of optically accessible exciton states is needed.

We thank the reviewer for the comment and we apologize for not being clear enough in explaining our theoretical model. We are not claiming to resolve the in-plane momentum of all the excitonic states and consequently the dark excitons lying outside the light cone. Our visible/near IR probe pulses can only couple to excitons within the light cone. However, as reported in the reply to the first reviewer, the dynamics of bright excitons are indirectly determined by the dynamics and scattering of dark excitons with finite momentum even if the electron or the hole part of their excitonic wavefunctions lies outside the light cone. In the specific case of the HS, bright excitons (such as A_W and $K-K$ ILX) are perturbed or "bleached" by Pauli blocking processes because they share the electron or the hole part of dark excitonic states (see equations 6-8). The strength with which hot excitons bleach bright exciton states and affect their dynamics depends on the momentum spread of the probed exciton transitions (via the Pauli blocking weighting factors).

ACTION TAKEN: We have added the following sentences on page 7 in the manuscript:

We stress that in our theoretical model, ILX with finite momenta Q can contribute via Pauli blocking process to the DTS signals calculated at the energies of bright (i.e. momentum direct) excitonic transitions, even if they lie outside the light cone (see eq. 2-3 in Methods section). This is due to the fact that bright excitons share their electron or hole part with finite momenta excitonic states.

2. Is the long formation time unique to hole-transfer in WSe₂/MoSe₂ bilayers? A small additional measurement that the authors could perform (with existing samples) is exciting the WSe₂ and looking at electron charge transfer which may show different formation times due to scatter with the other conduction band minimum. Even though MoSe₂ is a smaller bandgap, it has very little absorption at the WSe₂ resonance.

We thank the reviewer for their comment. We point out that in this work, we chose to measure the transient absorption (TA) response upon selective photoexcitation of MoSe₂ A exciton transition in order to reduce the number of possible scattering processes occurring in the HS. Following the reviewer's suggestion, we have performed additional measurements of the broadband non-equilibrium optical response of the HS upon photoexcitation on resonance to the WSe₂ A exciton at $\hbar\omega = 1.7$ eV. In this condition, the photoexcitation results in a direct formation of an exciton population in both the layers [9, 10], because the absorption of MoSe₂ at the WSe₂ resonance is not negligible. Also in this case, we found a clear PB signal of the ILX characterized by similar build up time as the one measured for A_{Mo} resonant excitation.

ACTION TAKEN: We have performed additional measurements on the HS by resonantly pumping the A_W exciton at $\hbar\omega = 1.70$ eV. A figure with this data (Fig R2) has been added to the SI section 2.2 with the following discussion.

A_W resonant excitation We have performed additional measurements of the ILX dynamics upon resonant excitation of the A_W exciton at $\hbar\omega_{pump} = 1.70$ eV (Supplementary Fig. 10). Similarly to resonant excitation of A_{Mo} (Fig. 2), the resonantly pumped A_W exciton dynamics are characterized by pulse-width limited rise while the A_{Mo} exciton shows a delayed bleaching signature due to interlayer electron transfer (Supplementary Fig. 10 red). The ILX signatures (Supplementary Fig. 10 green) show similar delayed bleaching dynamics as upon resonant excitation of A_{Mo} (Fig. 2), though with different rise times. We note that with resonant excitation of the A_W exciton at $\hbar\omega = 1.70$ eV, it is difficult to disentangle the effect of interlayer electron transfer on the ILX formation time from excitation of hot A_{Mo} excitons [9, 10], followed by interlayer hole transfer and subsequent ILX formation.

Fig. R 2 Exciton dynamics upon A_W resonant excitation The dynamics of the A_{Mo} (red), A_W (purple), and ILX (green) are shown for resonant photoexcitation of the A_W exciton at 77 K with pump fluences of $13 \mu\text{J}/\text{cm}^2$. Dots represent raw data.

Medium concerns:

3. On page 3, paragraph 2, the authors attribute the broad ILX emission peak (in Figure 1c) to the contribution of both momentum-direct and momentum-indirect transitions. If this were a significant contribution, we would expect the absorption to be much narrower, which does not appear to be the case in Figure 2b. It seems more likely that they are probing sample inhomogeneity.

We thank the reviewer for their comment. As discussed in response to Comment #1, the broadening in the linear absorption (Fig. 1b) and TA signatures (Fig. 2) are not due the momentum indirect excitons but are due to disorder, as the Reviewer states. We agree that the signal we report in Fig. 2b is broadened due to sample inhomogeneity. Given our large spot size in the TA experiment (100 microns) we are effectively performing an ensemble measurement which greatly contributes to the signal width.

ACTION TAKEN: We have added a phrase to the manuscript on pg. 3 to explicitly state that our measurements include inhomogeneous broadening due to our beam spot diameters:

The pump and probe are focused to spot diameters on the order of 100 μm , such that sample heterogeneity contributes to inhomogeneous broadening of the TA peaks.

4. In Figure 1 c, the power density and wavelength used to generate the photoluminescence are not reported.

ACTION TAKEN: We have updated the caption of Fig. 1 to include the requested power density and wavelength, included below for reference:

c, Linear absorption (black trace) and PL (green trace) spectra of the HS at 77 K. The PL spectrum was measured in a confocal Raman microscope (inVia, Renishaw) using 530 nm continuous wave excitation at $9 \times 10^7 \mu\text{W}/\text{cm}^2$.

5. On page 4, paragraph 2, the authors attribute the blueshift from the absorption to the emission to the energy difference between momentum-direct and -indirect transitions. While this is a possible explanation, the sample inhomogeneity can explain the Stark shift more simply. Exciton diffusion, particularly of hot excitons, allows the excitons to find the energetic minima in the potential landscape whereas absorption will probe the states available and thus on average will probe a higher energy.

We thank the reviewer for their comment. We are not reporting a blue shift of the ILX in Fig. 2b but rather our explanation given on page 4, paragraph 2 is that our optical measurement will only probe the optically bright K-K exciton whereas the PL measurement reports on the K-K and other momentum indirect ILX excitons [11]. We agree that exciton-phonon scattering due to defects will result in broadening of our reported linewidths. Moreover the timescale of hot exciton diffusion has been estimated to be much slower than sub-ps build-up dynamics observed in our experiments [12].

Minor concerns:

6. The authors refer to the indirect transition as one between K - Σ as in the conduction band minima is at Σ . Given the plethora of notations of this point of the Brillouin zone (which sometimes called Q / Λ in papers such as Hanbicki et al. ACS Nano, 12(5) 2018 and Rivera et al. Nature Nanotech 13, 2018), it would be useful to define its position in momentum space as you introduce it.

ACTION TAKEN: We thank the reviewer for this comment. We realize that there are several conventions for the nomenclature of this point in the Brillouin zone and have added the following clarifying statement on page 3 of the main text:

Both momentum-direct ($K - K$) and momentum-indirect ($\Sigma - K$), where Σ has also been referred to as the Q/Λ point [13–15].

7. The dashed/dot-dashed lines in Fig. 3f,g, and h are visually hard to distinguish from one another.

We thank the reviewer for bringing the legibility to our attention. We have modified the dashing and line thickness to hopefully enhance the legibility of Fig. 3f-h (included below).

Reviewer #3 (Remarks to the Author):

In this manuscript, the authors studied the interlayer exciton formation dynamic process. They found that the interlayer excitons formation timescale is nearly an order of magnitude (~ 1 ps) longer than the interlayer charge transfer time (~ 100 fs). They attributed the relative delay to an interplay between a phonon-assisted interlayer exciton cascade and subsequent cooling processes. The results are important for exciton physics. Although this work has been carried out with care, some issues still exist. The interlayer exciton formation dynamics are really difficult to access directly due to the weak oscillator strength and the ultrafast interlayer charge transfer process. However, in this manuscript, the signal strength $\Delta T/T$ is very very weak in the range of interlayer exciton. The signal needs to be amplified a hundred times to see the outline. Moreover, photobleaching signal peaks generally increase on the picosecond scale (Nature Photonics 2020,14, 171-176; 2021,15, 238–244). Further proof is needed that this is not caused by noise signal.

We thank the reviewer for their comments and for judging our results important for exciton physics. We have considered their concerns about our signal assignment of the ILX. We would like to point out that the references mentioned by the Reviewer (Nature Photonics 2020,14, 171-176; 2021,15, 238–24) regard studies of colloidal quantum dots and perovskite-based materials, respectively, and exhibit different exciton dynamics than the TMD HS system studied here.

Regarding the signal strength of the ILX, we note that though the ILX signal is 100 time weaker than the intralayer signatures, as noted by the reviewer and shown in Fig. 2b, this signal is still well above our noise floor which can be seen clearly by comparing the black spectrum for probe delay of -0.3 ps with the positive delay times in Fig. 2b. We have performed a number of measurements to ensure that the signals we assign to the ILX cannot be attributed to either systematic noise nor to other signals from the TMD HS or constituent MLs. We have observed the ILX signature with very similar signal strengths and dynamics using two separate spectroscopic instruments with very different fundamental lasers and detection schemes (See Methods: Transient Absorption Spectroscopy), eliminating the possibility that the ILX signature is the result of a systematic signal of the spectrometer. We have measured the ILX signal as a function of fluence (Supplementary Fig. S9), pump photon energy (see response to Reviewer #2, concern #2 resulting in Fig. R2), and temperature (to Reviewer #3 response below below, Fig. R3) and see that the ILX signature persists. We also report the circular dichroism (CD) signal of the ILX in Supplementary Fig. S16; the CD of the ILX has been previously reported with PL measurements [14]. We have additionally performed control experiments on the constituent monolayers (Supplementary Fig. S13 & S14) where we see no signatures similar to what we assign to be the ILX.

The relaxation process of hot interlayer exciton should take different times at different temperatures. As the authors cite the literature 24 (Nano Lett. 2021, 21, 2165-2173), the authors can consider other temperatures to confirm further the reliability of this signal, not just 77 K.

As discussed above, the signature of the ILX is 100 times weaker than the intralayer exciton at 77K and is

expected to further decrease in strength with increasing temperature. This phenomenon has been clearly documented in PL studies of several heterostructures, where the ILX peak is also seen to red shift with increasing temperature [14, 16]. To address the Reviewer's question directly, we have performed additional TA experiments at 100 K, 150 K, and 290 K (Fig. R3). With increasing temperature, we see a steady decrease in the ILX signature up to room temperature. The formation time of the ILX does not change with temperature in agreement with previous measurements showing that interlayer charge transfer (which determines the ILX formation) is temperature independent (see ref. 24 of the main text).

ACTION TAKEN: We have added the following figure showing the temperature dependence of the ILX dynamics and signal strength to the Supplementary Information Section 2.2 along with the following descriptive text:

ILX Temperature Dependence We have performed additional TA experiments at 100 K, 150 K, and 290 K (Supplementary Fig. 3). We see a steady decrease of the ILX PB signal with increasing temperature, up to room temperature in agreement with the quench of the ILX oscillator strength previously observed in static PL measurements [16]. The build dynamics of the PB signal of the ILX do not substantially change at high temperatures, in agreement with previous pump-probe optical measurements showing that interlayer charge transfer process is temperature independent [17].

Fig. R 3 Temperature dependence of the ILX The ILX dynamics (a), and peak signal strength (b, orange) are shown for 4° HS at 77, 140, 210, and 290 K. As the temperature increases, the ILX exciton signal strength decreases (b). Error bars indicate the standard deviation of the signal amplitude around the signal maximum.

References

- [1] Zhu, H. *et al.* Interfacial Charge Transfer Circumventing Momentum Mismatch at Two-Dimensional van der Waals Heterojunctions. *Nano Letters* **17** (6), 3591–3598 (2017) .
- [2] Zimmermann, J. E. *et al.* Ultrafast Charge-Transfer Dynamics in Twisted MoS₂/WSe₂ Heterostructures. *ACS Nano* **15** (9), 14725–14731 (2021) .
- [3] Holler, J. *et al.* Interlayer exciton valley polarization dynamics in large magnetic fields. *Physical Review B* **105** (8), 1–9 (2021) .
- [4] Hagel, J., Brem, S., Linderälv, C., Erhart, P. & Malic, E. Exciton landscape in van der Waals heterostructures. *Physical Review Research* **3** (4), 043217 (2021) .
- [5] Meneghini, G., Reutzler, M., Mathias, S., Brem, S. & Malic, E. Direct visualization of hybrid excitons in van der Waals heterostructures. *Preprint* 1–7 (2023) .
- [6] Jin, Z., Li, X., Mullen, J. T. & Kim, K. W. Intrinsic transport properties of electrons and holes in monolayer transition-metal dichalcogenides. *Physical Review B - Condensed Matter and Materials Physics* **90** (4), 1–7 (2014) .
- [7] Meneghini, G., Brem, S. & Malic, E. Ultrafast phonon-driven charge transfer in van der Waals heterostructures. *Natural Sciences* **2** (4), 1–7 (2022) .
- [8] Mahdikhanyarvejahany, F. *et al.* Localized interlayer excitons in MoSe₂–WSe₂ heterostructures without a moiré potential. *Nature Communications* **13** (1), 1–6 (2022) .
- [9] Policht, V. R. *et al.* Dissecting Interlayer Hole and Electron Transfer in Transition Metal Dichalcogenide Heterostructures via Two-Dimensional Electronic Spectroscopy. *Nano Letters* **21** (11), 4738–4743 (2021) .
- [10] Trovatiello, C. *et al.* The ultrafast onset of exciton formation in 2D semiconductors. *Nature Communications* **11** (1), 5277 (2020) .
- [11] Okada, M. *et al.* Direct and Indirect Interlayer Excitons in a van der Waals Heterostructure of hBN/WS₂/MoS₂/hBN. *ACS Nano* **12** (3), 2498–2505 (2018) .
- [12] Rosati, R. *et al.* Non-equilibrium diffusion of dark excitons in atomically thin semiconductors. *Nanoscale* **13** (47), 19966–19972 (2021) .
- [13] Kormányos, A. *et al.* $k \cdot p$ theory for two-dimensional transition metal dichalcogenide semiconductors. *2D Materials* **2** (2), 022001 (2015) .

- [14] Hanbicki, A. T. *et al.* Double Indirect Interlayer Exciton in a MoSe₂/WSe₂ van der Waals Heterostructure. *ACS Nano* **12** (5), 4719–4726 (2018) .
- [15] Rivera, P. *et al.* Interlayer valley excitons in heterobilayers of transition metal dichalcogenides. *Nature Nanotechnology* **13** (11), 1004–1015 (2018) .
- [16] Nagler, P. *et al.* Interlayer exciton dynamics in a dichalcogenide monolayer heterostructure. *2D Materials* **4** (2), 025112 (2017) .
- [17] Wang, Z. *et al.* Phonon-Mediated Interlayer Charge Separation and Recombination in a MoSe₂/WSe₂ Heterostructure. *Nano Letters* **21** (5), 2165–2173 (2021) .

Reviewers' Comments:

Reviewer #1:

Remarks to the Author:

I would like to thank the authors very much for addressing all my concerns. From my perspective, the manuscript is now ready for publication in Nature Communications.

Reviewer #2:

Remarks to the Author:

Overall, the experimental results and theoretical simulations of the paper are of interest and merit. Many of the original issues that the other reviewers and I have brought up were dealt with rather successfully. However, the writing and discussions in the paper leave much to be desired. More specifically:

1) The Theoretical Model section needs to be simplified, made more cohesive and complete.

Without a coherent story, it's hard to get a good understanding of the model/physics going on.

Here are a few examples:

1a. The authors provide the same identical explanation regarding the contribution of finite momentum ILXs to the DTS twice; once on page 5 and once on page 7.

1b. It's mentioned that scattering to Σ valley needs to be considered for the model and refers the reader to the SI. They never report the $\Delta\Sigma$ used for the figures in the main text. Also, it's never mentioned whether $\Delta\Sigma = \Sigma-K$ or $\Delta\Sigma = K-\Sigma$.

1c. Excitons are called fermions on page 7. They are quasi-bosons. I believe the authors mean to say the fermionic nature of holes and/or electrons.

1d. On page 7, the authors mention that the PB dynamics should not be dependent on twist-angle. However, $\Delta\Sigma$ is a function of layer hybridization which, in principle, should be dependent on twist-angle.

1e. In relation to the last comment (1d): References 28 and 51 are cited incorrectly on page 7 to back up a claim that the charge transfer time is mostly independent of twist-angle.

1e-i) Ref. 28 finds that the charge transfer time can change by 5x

1e-ii) Ref. 51 cannot resolve the charge transfer time due to poor temporal resolution.

1e-iii) Additionally, Ref 28 and 51 took measurements in MoS₂/WSe₂ heterostructures which do not have the complication of a nearby Σ valley (see comment 1d).

2) The authors explain the nomenclature of Σ on page 3 but first use it on page 2.

3) Page 2: 4° is not nearly-aligned (57° is also not nearly anti-aligned). I would not use the term nearly-(anti)aligned for anything further than 1° from either 0° (60°). Instead, say something like "close-to H / R stacked" or "near" or something similarly more accurate.

4) Supplementary Figure 10 shows fits without the fitting parameters (i.e. the apparent ILX formation time which is stated to be similar to that of the main-text).

5) The authors neglect to mention how they obtain the estimated exciton density on page 4. This could go in the methods or the supplementary information.

Reviewer #3:

Remarks to the Author:

Exploring the interlayer exciton formation dynamics is important but very difficult. For some of the questions raised earlier, although the authors gave corresponding explanations and supplementary experiments, there are still some issues to be addressed.

1, For this 100-fold enlarged $\Delta T/T$ spectra, repeatability still needs to be verified. Moreover, for heterojunction with an interlayer twist angle of 57°, these fitted lines are misleading in Figure S17, they have no apparent regularity.

2, It is expected that the interlayer charge transfer time is independent of temperature. The

process is fast around 100 fs. However, ILX formation timescale (delay to ps scale) is also independent of temperature. The experimental result contradicts the author's opinion. As explained in abstract, microscopic calculations attribute the relative delay to an interplay between a phonon-assisted interlayer exciton cascade and subsequent cooling processes, and excitonic wave-function overlap. The phonon-assisted process and subsequent cooling processes should be directly related to temperature. As the temperature increases, the formation process may be faster according the author's explanation.

Reviewer Reply: Time-domain observation of interlayer exciton formation and thermalization in a MoSe₂/WSe₂ heterostructure

REVIEWER COMMENTS

Reviewer #1 (Remarks to the Author):

I would like to thank the authors very much for addressing all my concerns. From my perspective, the manuscript is now ready for publication in Nature Communications.

We thank the reviewer for their positive comments and opinion that our revisions adequately prepared our manuscript for publication.

Reviewer #2 (Remarks to the Author):

Overall, the experimental results and theoretical simulations of the paper are of interest and merit. Many of the original issues that the other reviewers and I have brought up were dealt with rather successfully. However, the writing and discussions in the paper leave much to be desired. More specifically:

We thank the reviewer for their positive comments on the interest and merit of our manuscript. We greatly appreciate the careful read the reviewer has done of our manuscript and have endeavoured to address the raised concerns.

1) The Theoretical Model section needs to be simplified, made more cohesive and complete. Without a coherent story, it's hard to get a good understanding of the model/physics going on. Here are a few examples:

We thank the reviewer for the thorough reading of our manuscript. We discuss all the raised points 1a-e in detail and in addition we rewrite parts of the theory section to discuss the interpretation in a more coherent fashion.

ACTION TAKEN: We rewrote a substantial amount of the Theoretical Model part in the main part of the manuscript on pages 5 - 8 (please see the marked-up manuscript for changes in red), as well as some minor parts in the Methods section regarding the theory. We hope everything is more coherent and clear now. All changes are marked in red. We have also streamlined the notation in the entire manuscript.

1a. The authors provide the same identical explanation regarding the contribution of finite momentum ILXs to the DTS twice; once on page 5 and once on page 7.

We thank the reviewer for identifying a redundant argument in our manuscript.

ACTION TAKEN: We deleted the doubled explanation on page 7.

1b. It's mentioned that scattering to Σ valley needs to be considered for the model and refers the reader to the SI. They never report the $\Delta\Sigma$ used for the figures in the main text. Also, it's never mentioned whether $\Delta\Sigma = \Sigma-K$ or $\Delta\Sigma = K-\Sigma$.

We thank the reviewer for identifying the accidentally omitted information on the $\Delta\Sigma$ value. We define $\Delta\Sigma$ as $E_{\Sigma} - E_K$, i.e. the difference between the energetic position E_K of the K valley and the energetic position E_{Σ} of the Σ valley. The value of $\Delta\Sigma$ which best reproduces our data is 5 meV.

ACTION TAKEN: We have added this to the main text on page 5:

(see Supplementary Note 1.4)

Also, we added this sentence to the Supplementary 1.4:

For the calculations in the main part of the manuscript, we use $\Delta\Sigma = E_{\Sigma} - E_K = 5 \text{ meV}$, where $E_{\Sigma/K}$ is the energetic position of the Σ/K valley, respectively.

1c. Excitons are called fermions on page 7. They are quasi-bosons. I believe the authors mean to say the fermionic nature of holes and/or electrons.

We thank the reviewer for pointing out this error; the reviewer is correct in that we intended to say 'fermionic' in

reference to the individual electrons and holes of the exciton.

ACTION TAKEN: The text has been modified as follows:

Because of the ~~fermionic~~ fermionic character of electron and hole forming the exciton as a quasi-boson, finite COM excitons lead ...

1d. On page 7, the authors mention that the PB dynamics should not be dependent on twist-angle. However, $\Delta\Sigma$ is a function of layer hybridization which, in principle, should be dependent on twist-angle.

We agree with the reviewer that interlayer twist angle will impact band hybridization. When we wrote that the PB dynamics are weakly dependent on the twist angle, we meant that this is expected to happen only for small rotational mismatch with respect to the perfectly aligned or anti-aligned stacking. We apologize to the reviewer for not being clear enough in our previous statement.

ACTION TAKEN: We have changed the sentence on page 6, which now reads as follows:

...result in PB DTS dynamics that are only weakly dependent on the twist angle at the relatively small angles deviating from strictly aligned and anti-aligned stacking [11].

1e. In relation to the last comment (1d): References 28 and 51 are cited incorrectly on page 7 to back up a claim that the charge transfer time is mostly independent of twist-angle. 1e-i) Ref. 28 finds that the charge transfer time can change by 5x 1e-ii) Ref. 51 cannot resolve the charge transfer time due to poor temporal resolution. 1e-iii) Additionally, Ref 28 and 51 took measurements in MoS₂/WSe₂ heterostructures which do not have the complication of a nearby Σ valley (see comment 1d).

We thank the reviewer for these precise comments and indeed in the previous version of the manuscript, not all references were appropriately chosen.

1e-i) We agree with the Reviewer that Ref 28 ([1]) is an inappropriate choice for supporting our argument of interlayer twist-angle independent ICT and note that the angles investigated in this study are greater than those studied in this manuscript. We had intended to cite Ji, et al. [2], which is now done.

1e-ii) While Ref. 51 ([3]) cannot resolve the charge transfer time as a function of the interlayer twist angle because it occurs faster than the instrument response time, the quoted experimental time resolution is <40 fs, which is significantly smaller than the time resolution of our TA measurements of 140 fs, and is on the order of ICT as measured using Two-dimensional Electronic spectroscopy of MoS₂/WS₂ with an instrument response function of 22 fs [4].

1e-iii) The reviewer is correct that the system studied in Ref.s 28 and 51 (MoS₂/WSe₂) is different than that studied here (MoSe₂/WSe₂). However, the Σ -point hybridization has also been shown in bandgap calculations of other heterostructures, i.e. WS₂/WSe₂ [5] and MoS₂/WS₂ [6].

To summarize, what we can conclude from literature is the following: (i) ICT timescale depends on the twist angle but sizeable variation of the dynamics are expected only for large orientational mismatch between the layers and (ii) in order to detect the variation of the ICT timescale, a temporal resolution of few tens of fs (i.e. much higher than that of the present experiments) is needed. ACTION TAKEN: We have modified the references on page 8 to remove the erroneous reference to ref. 28 and have replaced it with Ji, et al. [2], which is now reference number 50.

...[~~49~~28, 50].

2) The authors explain the nomenclature of Σ on page 3 but first use it on page 2.

We thank the reviewer for their comment.

ACTION TAKEN: We have moved the comment on the definition of the Σ point from page 3 to follow its introduction on page 2:

A recent tr-ARPES study [7] was able to track the ILX formation process following a phonon-assisted interlayer electron transfer as mediated by intermediate scattering to the Σ valleys, where Σ has also been referred to as the Q/ Λ point [12,37,38]

3) Page 2: 4° is not nearly-aligned (57° is also not nearly anti-aligned). I would not use the term nearly-(anti)aligned for anything further than 1° from either 0° (60°). Instead, say something like “close-to H / R stacked” or “near” or something similarly more accurate.

We appreciate the reviewer’s comment on our language for the degree of alignment and they raise a valid point.

ACTION TAKEN: We have adopted the reviewer’s suggested terminology of “near” throughout the manuscript and supplementary information when discussing the experimental data, in particular on pages 3 & 8 of the main text.

...nearly- near ...

4) Supplementary Figure 10 shows fits without the fitting parameters (i.e. the apparent ILX formation time which is stated to be similar to that of the main-text).

ACTION TAKEN: We have added the fitting parameters of the ILX rise time upon resonant excitation of the A_W in the caption for Supplementary Figure 10, and have additionally added the multiplicative factors for normalization to the figure legend (Fig. R1):

Fig. R 1 Exciton dynamics upon A_W resonant excitation The dynamics of the A_{M_o} (red), A_W (purple), and ILX (green) are shown for resonant pumping of the A_W exciton at 77 K with pump fluences of $13 \mu J/cm^2$. Dots represent raw data with fits shown as solid lines. The rise of the A_{M_o} was fit to 100 fs and the ILX rise time was fit to 320 fs.

5) The authors neglect to mention how they obtain the estimated exciton density on page 4. This could go in the methods or the supplementary information.

We thank the reviewer for this comment.

ACTION TAKEN: We have added the details of the exciton density calculation to the Methods section on pg. 8, and we have modified Supplementary Fig. 2 (Fig. R2) to include the units of linear absorption for the TMD HS spectrum.

The exciton density of $\approx 3.6 \times 10^{11} \text{ cm}^{-2}$ is calculated using the pump fluence of $2.8 \mu\text{J cm}^{-2}$, the pump spectrum and the linear absorption spectrum of the TMD HS at 77 K (Supplementary Fig. 2).

We have added a reference to this information to the main text on pg. 4

...($\approx 3.6 \times 10^{11} \text{ cm}^{-2}$, see Methods for details)...

Fig. R 2 Experimental Spectra and Setup. **a**, Normalized pump (yellow) and probe (blue) pulse spectra shown with the 77 K static linear absorption spectrum of the HS. **b**, Spectrally resolved transient absorption setup. BS – beam splitter; CM – curved mirror, WLC – YAG white light crystal; SPF - Short Pass Filter.

Reviewer #3 (Remarks to the Author):

Exploring the interlayer exciton formation dynamics is important but very difficult. For some of the questions raised earlier, although the authors gave corresponding explanations and supplementary experiments, there are still some issues to be addressed.

We thank the reviewer for their consideration of our manuscript.

1, For this 100-fold enlarged $\Delta T/T$ spectra, repeatability still needs to be verified. Moreover, for heterojunction with an interlayer twist angle of 57° , these fitted lines are misleading in Figure S17, they have no apparent regularity.

We thank the reviewer for their comment and for identifying an error in the caption of Supplementary Fig. 17. With regard to the "fitted lines in Figure S17," the solid lines in panel b are not fits but a smoothed representation of the data, as is the solid line in panel c. We would like to bring Supplementary Fig. 9a to the reviewer's attention (included below, Fig. R3) which presents the ILX signature of Supplementary Fig 17 as a two-dimensional map in frequency and time, where the rise of the ILX signature is perhaps more clear.

ACTION TAKEN: We have modified the captions for Figure 2 and Supplementary Fig. 17 to clarify the presented data.

Figure S17...b, Select $\Delta T/T$ spectra from (a) at early time delays. The region below the A_{Mo} resonance is multiplied by a factor of 100 to highlight the weak ILX peak in the near-IR. **Solid lines in the near-IR region are a smoothed representation of the raw data in dots.** **c,** Temporal dynamics of the A_{Mo} (red, $\hbar\omega = 1.57$ eV), A_W (purple, $\hbar\omega = 1.70$ eV), and ILX (green, $\hbar\omega = 1.37$ eV) in the first 2 ps. The A_W and ILX peaks are multiplied by factors of 3 and 290, respectively, to emphasize the delayed rise. ~~Dashed/Dotted lines are the fits to the data.~~ **Green dashed/solid lines represent the raw and smoothed ILX signature, respectively...**

Fig. 2...b, $\Delta T/T$ spectra from (a) at selected early time delays. The region below the intralayer exciton optical gap is multiplied by a factor of 100 to highlight the weak ILX peak in the near IR. **Solid lines in the near-IR region are a smoothed representation of the raw data in dots...**

Regarding the veracity of ILX signature, we respectfully disagree that additional measurements need be performed to verify the repeatability of our measurements. As described in the manuscript, SI, and our response to the reviewer's previous round of comments, we have demonstrated that the weak ILX signatures are highly repeatable by the series of measurements reiterated below.

- Measurements of the ILX signature using the same transient absorption instrument under a) different pump wavelength conditions (Supplementary Fig. 10, b) pump fluences (Supplementary Fig. 9), and c) temperature conditions (Supplementary Fig. 11),
- Measurements of the ILX signature using a second, completely independent, transient absorption setup under similar temperature, pump fluence, and pump wavelength conditions (Fig. 2 & Supplementary Fig. 8),
- Measurement of an ILX signature in two separate HS samples with 4° (Fig. 2) and 57° (Supplementary Fig. 17) interlayer twist angles,
- Measurement of circular dichroism of the ILX signature (Supplementary Fig. 16),
- Control measurements on the constituent monolayers, WSe₂ and MoSe₂ (Supplementary Figs 13 & 14).

Fig. R 3 Fluence dependent ILX dynamics. 2D $\Delta T/T$ maps of the ILX region with pump fluences of $5 \mu\text{J}/\text{cm}^2$ (a) and $50 \mu\text{J}/\text{cm}^2$ (b). c, $\Delta T/T$ spectra at 4 ps from a and b. As pump fluence increases the positive ILX PB signal is obscured by the growing negative PA signal from the A_{M_o} exciton. Dashed lines represent the full signal with the smoothed data as solid lines.

We greatly appreciate that the ILX signatures measured are weak compared to the intralayer peaks (100x weaker) and have therefore performed the above listed measurements, and more, to improve our certainty of our assignment. Taken together, these measurements provide, in our opinion, compelling evidence of the presence of a PB signal from the ILX.

2, It is expected that the interlayer charge transfer time is independent of temperature. The process is fast around 100 fs. However, ILX formation timescale (delay to ps scale) is also independent of temperature. The experimental result contradicts the author's opinion. As explained in abstract, microscopic calculations attribute the relative delay to an interplay between a phonon-assisted interlayer exciton cascade and subsequent cooling processes, and excitonic wave-function overlap. The phonon-assisted process and subsequent cooling processes should be directly related to temperature. As the temperature increases, the formation process may be faster according to the author's explanation.

We thank the reviewer for the comment on the expected behavior of the ILX formation time with temperature. To better comment on the temperature dependent experimental results in Fig. R5 (Supplementary Fig. 12), we have performed additional calculations of the ILX DTS at 150 K and 300 K (Fig. R4). These calculations show that the DTS of the ILX displays faster build up time at increasing temperatures, as proposed by the referee, but the temperature effect is extremely weak. This happens because hot ILX relaxation process is mainly governed by spontaneous phonon emission processes which do not depend on the temperature (see the electron-phonon

scattering rate reported in Eq. S15). Since the differences in the observed build-up times are within the signal-to-noise ratio of the experiment, we are unable to make a strong conclusion about the ILX temperature dependence.

ACTION TAKEN: We have performed additional calculations of DTS traces of ILX at 150 K and 300 K (Fig. R4). These calculations have been added to Section 2.3 of the Supplementary Information, as figure S.11. Figure S.12 has been modified with respect to the previous version: a panel with normalized temporal traces has been added. All the Section 2.3 has been rewritten as follows:

2.3 Temperature Dependent ILX dynamics

We have performed DTS calculations (Supplementary Fig. 4) and TA measurements (Supplementary Fig. 5) of the ILX dynamics at increasing values of the temperature. The rise dynamics of the calculated DTS, weakly depends on the temperature. The modest increase of the ILX buildup time at higher temperatures is due to the fact that hot ILX population relaxes to the ground state by emitting phonons. Eq. S15, describes the phonon-assisted electron and hole scattering rate. In this equation, phonon emission and absorption processes are depicted by the factors $(n + \frac{1}{2} \pm \frac{1}{2})$, where "+" denotes emission and "-" denotes absorption. If we consider phonon emission, the term reads $(n + 1)$. Here, the part of the equation with the factor equal to n describes the stimulated emission process while the part with the factor equal to 1 describes the spontaneous emission process. The latter process does not depend on the temperature. The experimental $\Delta T/T$ traces display a steady decrease of the intensity with increasing temperature up to $T=300$ K, in agreement with the quench of the ILX oscillator strength previously observed in static PL measurements [8]. The timescale of the build up dynamics seems not to change within the explored temperature range. Since the differences in the observed build-up times are within the signal-to-noise ratio of the experiment, we are unable to make a strong conclusion about the temperature dependence of this process.

Fig. R 4 Temperature Dependence of the DTS traces of ILX DTS signals of ILX are calculated at 77 K (green), 150 K (yellow), and 300 K (red). The buildup time is slightly faster at increasing temperature.

Fig. R 5 Temperature dependence of the ILX dynamics (a) ILX PB dynamics at increasing temperatures . (b) Normalized traces. (c) $\Delta T/T$ peak signal strength as a function of the temperature. The error bars indicate the standard deviation of the signal amplitude around the signal maximum.

References

- [1] Zimmermann, J. E. *et al.* Ultrafast Charge-Transfer Dynamics in Twisted MoS₂/WSe₂ Heterostructures. *ACS Nano* **15** (9), 14725–14731 (2021) .
- [2] Ji, Z. *et al.* Robust Stacking-Independent Ultrafast Charge Transfer in MoS₂/WS₂ Bilayers. *ACS Nano* **11** (12), 12020–12026 (2017) .
- [3] Zhu, H. *et al.* Interfacial Charge Transfer Circumventing Momentum Mismatch at Two-Dimensional van der Waals Heterojunctions. *Nano Letters* **17** (6), 3591–3598 (2017) .
- [4] Policht, V. R. *et al.* Dissecting Interlayer Hole and Electron Transfer in Transition Metal Dichalcogenide Heterostructures via Two-Dimensional Electronic Spectroscopy. *Nano Letters* **21** (11), 4738–4743 (2021) .
- [5] Wu, K. *et al.* Identification of twist-angle-dependent excitons in WS₂/WSe₂ heterobilayers. *National Science Review* **9** (6) (2022) .
- [6] Okada, M. *et al.* Direct and Indirect Interlayer Excitons in a van der Waals Heterostructure of hBN/WS₂/MoS₂/hBN. *ACS Nano* **12** (3), 2498–2505 (2018) .
- [7] Schmitt, D. *et al.* Formation of moiré interlayer excitons in space and time. *Nature* **608** (7923), 1–6 (2022) .
- [8] Nagler, P. *et al.* Interlayer exciton dynamics in a dichalcogenide monolayer heterostructure. *2D Materials* **4** (2), 025112 (2017) .

Reviewers' Comments:

Reviewer #3:

Remarks to the Author:

I would like to thank the authors very much for addressing all my concerns. From my perspective, the manuscript is now ready for publication in Nature Communications.

Reviewer Reply: Time-domain observation of interlayer exciton formation and thermalization in a MoSe₂/WSe₂ heterostructure

REVIEWER COMMENTS

Reviewer #3 (Remarks to the Author):

I would like to thank the authors very much for addressing all my concerns. From my perspective, the manuscript is now ready for publication in Nature Communications.

We thank the reviewer for their comments on our manuscript and for their opinion that the manuscript is now ready for publication in Nature Communications.